# Dichotomous Diffusion Policy Optimization

**Ruiming Liang**[1,2*‡]**, Yinan Zheng**[3*¶]**, Kexin Zheng**[4*‡]**, Tianyi Tan**[3*]**, Jianxiong Li**[3]**,**
**Liyuan Mao**[5]**, Zhihao Wang**[6‡]**, Guang Chen**[7]**, Hangjun Ye**[7]**, Jingjing Liu**[3]**,**
**Jinqiao Wang**[1,2†]**, Xianyuan Zhan**[3†]

[1] Fundation Model Research Center, Institute of Automation, Chinese Academy of Sciences
[2] School of Artificial Intelligence, University of Chinese Academy of Sciences
[3] Institute for AI Industry Research (AIR), Tsinghua University
[4] The Chinese University of Hong Kong     [5] Shanghai Jiao Tong University
[6] Peking University     [7] Xiaomi EV
`liangruiming2024@ia.ac.cn, zhengyn23@mails.tsinghua.edu.cn,`
`zhanxianyuan@air.tsinghua.edu.cn`

## Abstract

Diffusion-based policies have gained growing popularity in solving a wide range of decision-making tasks due to their superior expressiveness and controllable generation during inference. However, effectively training large diffusion policies using reinforcement learning (RL) remains challenging. Existing methods either suffer from unstable training due to directly maximizing value objectives, or face computational issues due to relying on crude Gaussian likelihood approximations, which require a large amount of sufficiently small denoising steps. In this work, we propose *DIPOLE* (**Di**chotomous diffusion **Pol**icy improv**e**ment), a novel RL algorithm designed for stable and controllable diffusion policy optimization. We begin by revisiting the KL-regularized objective in RL, which offers a desirable weighted regression objective for diffusion policy extraction, but often struggles to balance greediness and stability. We then formulate a greedified policy regularization scheme, which naturally enables decomposing the optimal policy into a pair of stably learned dichotomous policies: one aims at reward maximization, and the other focuses on reward minimization. Under such a design, optimized actions can be generated by linearly combining the scores of dichotomous policies during inference, thereby enabling flexible control over the level of greediness. Evaluations in offline and offline-to-online RL settings on ExORL and OGBench demonstrate the effectiveness of our approach. We also use *DIPOLE* to train a large vision-language-action (VLA) model for end-to-end autonomous driving (AD) and evaluate it on the large-scale real-world AD benchmark NAVSIM, highlighting its potential for complex real-world applications. Project Page: https://lrmbbj.github.io/DIPOLE/

## 1 Introduction

Due to the strong capability of diffusion models (Sohl-Dickstein et al., 2015; Ho et al., 2020) in modeling multi-modal action distributions and controllable generation during inference (Dhariwal & Nichol, 2021; Ho & Salimans, 2022), modeling policies using diffusion models has become a popular choice in solving complex decision-making tasks such as embodied robotics (Chi et al., 2023; Zheng et al., 2026) and autonomous driving (Zheng et al., 2025b; Tan et al., 2025; Li et al., 2026). Although proven to be effective in imitation learning-based settings, training large diffusion/flow matching policies that surpass data-level performance with reinforcement learning (RL) (Sutton et al., 1998) has remained an important yet challenging direction.

---

[*]Equal contribution.
[†]Corresponding authors.
[‡]Work done during internships at Institute of AI Industry Research (AIR), Tsinghua University.
[¶]Project lead.

Training diffusion policies with RL faces numerous challenges, most notably, learning stability and computation efficiency. A naïve approach to train diffusion policies with RL is to directly optimize the reward or value objective via gradient backpropagation through the multi-step denoising process (Xu et al., 2023b; Clark et al., 2023), which often suffers from noisy and unstable gradient updates, while also being extremely costly. To avoid this, some studies adopt a compromise by freezing the diffusion model and instead searching for optimized noises (Wagenmaker et al., 2025; Hansen-Estruch et al., 2023), a strategy often referred to as inference-time scaling (Ma et al., 2025b). However, these approaches rely heavily on well-pretrained diffusion policies and are fundamentally limited by their performance upper bound. Another explored direction is to adopt policy gradient methods (such as PPO (Schulman et al., 2017)) for diffusion policy optimization, which models the denoising process as a multi-step Markov decision process (MDP) and uses Gaussian approximations to compute the log-likelihood of intermediate denoising steps (Black et al., 2024b; Ren et al., 2025). However, the crude Gaussian-based approximation only provides reasonable likelihood information when adopting sufficiently small denoising steps, which inevitably results in large exploration spaces and prolonged training, making such methods difficult to scale and prone to approximation error accumulation in practice. Therefore, a critical research question arises: *Can we build a more effective and stable RL method for diffusion policy optimization?*

To answer this question, we turn our attention to the KL-regularized RL objective, which offers a nice, closed-form weighted regression objective for optimal policy extraction (Peng et al., 2019). We can thus optimize a diffusion policy by incorporating an exponential reward- or value-based weighting term, scaled by a temperature parameter, into the standard diffusion regression loss (Lee et al., 2023; Kang et al., 2023; Zheng et al., 2024). Although promising, this approach also suffers from several limitations. A fundamental issue is that weighted regression can only achieve greedy reward maximization when the temperature parameter is set to a large value, which easily leads to exploding loss and training instability. Moreover, the learning loss becomes dominated by a small number of high-reward samples, which severely undermines training effectiveness and scalability even with increased data (Park et al., 2024). To address the previous challenges, we propose *DIPOLE* (**Di**chotomous diffusion **Pol**icy improv**e**ment), a novel RL framework designed for highly stable and controllable diffusion policy optimization. Specifically, we introduce a greedified KL-regularized RL objective, which regularizes policy learning towards a value-reweighted reference policy. Interestingly, we show that the original unstable exponential weighting term in the optimal policy can be decomposed into two bounded smooth dichotomous terms. This naturally allows us to decompose the optimal policy into a pair of stably learned dichotomous policies: one aims at reward maximization and the other focuses on reward minimization. Moreover, the optimized policy can be recovered through a linear combination of the scores from both dichotomous policies, which closely aligns with the widely used classifier-free guidance mechanism in diffusion models (Ho & Salimans, 2022), enabling perfect controllability over the greediness of action generation.

Extensive experimental results demonstrate the effectiveness of *DIPOLE* across a wide range of locomotion and manipulation tasks in ExORL (Yarats et al., 2022) and OGBench (Park et al., 2025a) benchmarks, evaluated under both offline and offline-to-online RL settings. Furthermore, we scale our learning approach to a large vision-language-action (VLA) model and evaluate it on the large-scale real-world autonomous driving benchmark NAVSIM (Dauner et al., 2024), showcasing significant performance improvements over the pre-trained baseline. These results highlight the strong applicability of *DIPOLE* for complex, real-world decision-making scenarios.

## 2 PRELIMINARY

**Reinforcement learning.** We consider the RL problem presented as a Markov Decision Process (MDP), which is specified by a tuple $\mathcal{M} := (\mathcal{S}, \mathcal{A}, \mathcal{P}, r, \gamma)$. $\mathcal{S}$ and $\mathcal{A}$ represent the state and action space; $\mathcal{P} : \mathcal{S} \times \mathcal{A} \to \Delta(\mathcal{S})$ is transition dynamics; $r : \mathcal{S} \times \mathcal{A} \to \mathbb{R}$ is the reward function; and $\gamma \in (0, 1)$ is the discount factor. We aim to find a policy $\pi : \mathcal{S} \to \Delta(\mathcal{A})$ that maximizes the expected return: $\mathbb{E}_\pi \left[ \sum_{k=0}^\infty \gamma^k \cdot r(s_k, a_k) \right]$. We define the discounted visitation distribution as: $d^\pi(s) = (1-\gamma) \sum_{k=0}^\infty \gamma^t p(s_k = s \mid \pi)$, which measures how likely to encounter $s$ when interacting with the environment using policy $\pi$. We also consider a replay buffer $\mathcal{D} = \{s_i, a_i, r_i, s_i'\}_{i=1}^N$, which can be a static dataset in the offline setting or dynamically updated with new samples in the offline-to-online setting. The state-value function and action-value functions are defined as: $V^\pi(s) = \mathbb{E}_\pi \left[ \sum_{k=0}^\infty \gamma^k \cdot r(s_k, a_k) \mid s_0 = s \right]$ and $Q^\pi(s, a) = \mathbb{E}_\pi \left[ \sum_{k=0}^\infty \gamma^k \cdot r(s_k, a_k) \mid s_0 = s, a_0 = a \right]$,

and the advantage function is defined as $A^\pi(s, a) = Q^\pi(s, a) - V^\pi(s)$. Their optimal counterparts under the optimal policy $\pi^\star$ are denoted as $V^\star$, $Q^\star$, and $A^\star$.

**Diffusion/flow matching policies.** Diffusion and flow matching models have attracted significant attention due to their strong expressiveness in capturing multi-modal data distributions, making them popular policy classes for complex decision-making tasks such as robotics (Chi et al., 2023; Black et al., 2024a) and autonomous driving (Zheng et al., 2025b; Tan et al., 2025). The action generation can be formulated as a state-conditional generation problem in which a probability path transforms a source distribution (typically a standard Gaussian) into a target action distribution. A neural network $\epsilon_\theta$ is trained to predict the noise along the path using the objective over a given dataset $\mathcal{D}$:

$$\mathcal{L}_{\epsilon_\theta} = \mathbb{E}_{t \sim U[0,1], \epsilon \sim \mathcal{N}(\mathbf{0}, \mathbf{I}), (s,a) \sim \mathcal{D}} \left[ \|\epsilon - \epsilon_\theta(a_t, s, t)\|^2 \right], \tag{1}$$

where $a_t = \alpha_t a + \sigma_t \epsilon$ (we use the subscript $t$ to distinguish diffusion steps from MDP steps $k$), with $\alpha_t$ and $\sigma_t$ being predefined noise schedules commonly used in score-based diffusion models (Song et al., 2021) or flow matching models (Lipman et al., 2022). The multi-step diffusion process endows diffusion models with strong distribution-fitting capabilities. However, it also poses challenges for RL fine-tuning: gradient propagation through the entire diffusion process is costly and unstable; the exact likelihood computation with diffusion models is intractable, causing a series of problems when optimizing with existing policy gradient RL methods due to approximation error.

## 3 METHODS

In this section, we revisit the KL-regularized objective for diffusion policy optimization, revealing its strengths and limitations. We then introduce *DIPOLE*, a novel RL framework that decomposes the optimization problem into dichotomous policy learning objectives, thereby enabling stable training and greedy diffusion policy extraction.

### 3.1 KL-REGULARIZED OBJECTIVE IN RL

Reinforcement learning with KL regularization is a highly flexible framework that has been widely used in various RL settings, which constrains policy optimization to remain close to a reference policy $\mu$, and has the following general form:

$$\max_\pi \ \mathbb{E}_{s \sim d^\pi(s)} \left[ \mathbb{E}_{a \sim \pi(a|s)} [G(s, a)] - \frac{1}{\beta} D_{\mathrm{KL}}(\pi(\cdot|s) \| \mu(\cdot|s)) \right], \tag{2}$$

where $\beta > 0$ is the temperature parameter, and $D_{\mathrm{KL}}(p\|q) = \mathbb{E}_{x \sim p}[\log(p(x)/q(x))]$. $G(\cdot)$ is the evaluated return to be maximized, which can either be the reward function $r(s, a)$ as in single-step problems such as LLM RL fine-tuning (Korbak et al., 2022; Shao et al., 2024), or the action-value function $Q^\pi(s, a)$ or advantage function $A^\pi(s, a)$ as in standard multi-step settings. The specific choice of reference policy $\mu$ gives rise to different RL task settings. For example, setting $\mu$ to be the uniform distribution, we recover maximum entropy RL as in SAC (Haarnoja et al., 2018); setting $\mu$ to be the behavior policy in offline datasets $\mathcal{D}$, we obtain many offline RL algorithms (Wu et al., 2019; Xu et al., 2023a; Garg et al., 2023; Mao et al., 2024); lastly, setting $\mu$ to be a pre-trained policy $\pi_0$ or the recently updated policy $\pi_{k-1}$, it corresponds to offline-to-online fine-tuning scenarios (Nakamoto et al., 2023; Li et al., 2023) or trust-region style online policy optimization (Schulman et al., 2015).

The flexibility of the KL-regularized RL framework makes it an ideal choice for diffusion policy optimization. The best part is, it is known that the optimization objective in Eq. (2) also provides a closed-form solution for optimal policy $\pi^\star$ as follows (Nair et al., 2020):

$$\pi^\star(a \mid s) \propto \mu(a \mid s) \cdot \exp(\beta G(s, a)), \tag{3}$$

Intuitively, the optimal policy is a reweighted version of the reference policy $\mu$, in which actions with higher values are assigned greater probability density. As shown in several existing studies (Kang et al., 2023; Zheng et al., 2024), if given a pre-trained diffusion policy $\epsilon_\theta$ trained with Eq. (1) as the reference policy $\mu$, we can further optimize it with the weighted diffusion loss in Lemma 1 to extract the optimal diffusion policy $\epsilon^\star$.

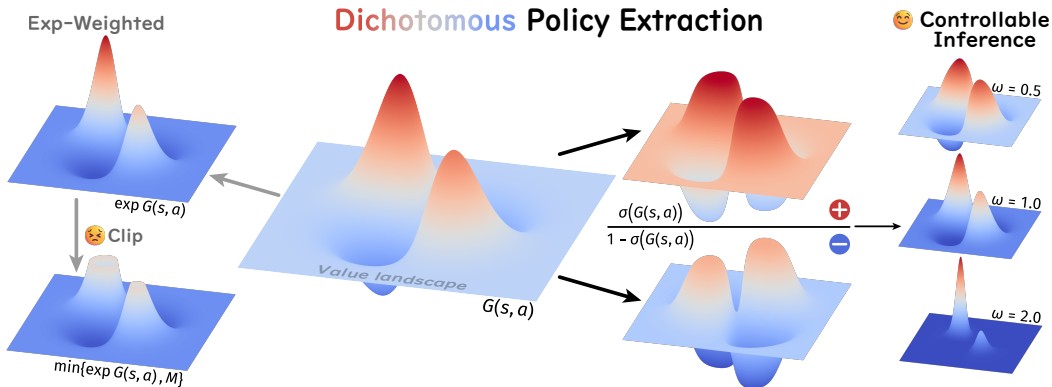

Figure 1: Illustration of the policy weighting scheme in *DIPOLE*. Based on our greedified policy optimization objective, the regression weight of the optimal policy can be decomposed into a pair of dichotomous terms, and the greediness for reward/value maximization can be flexibly controlled by $\omega$.

**Lemma 1.** *We can generate optimal $a \sim \pi^\star(a|s)$ in Eq. (3) by optimizing the weighted diffusion loss in Eq. (4) and solving the diffusion reverse process with obtained $\epsilon^\star$ (Zheng et al., 2024).*

$$\mathcal{L}_{\epsilon_\theta} = \mathbb{E}_{t\sim U[0,1],\epsilon\sim\mathcal{N}(\mathbf{0},\mathbf{I}),(s,a)\sim\mathcal{D}}\left[\exp(\beta G(s,a))\cdot\|\epsilon - \epsilon_\theta(a_t,s,t)\|^2\right]. \tag{4}$$

Compared to diffusion-based RL methods that rely on unstable reward/value maximization (Xu et al., 2023b; Clark et al., 2023) or biased likelihood approximation (Black et al., 2024b; Ren et al., 2025), Eq. (4) offers a simple and scalable training scheme for policy optimization, requiring only the addition of a weighted term to the base diffusion learning objective in Eq. (1). Despite its simplicity, we do not observe the adoption of this scheme in many recent diffusion-based RL methods. Why is that? Actually, there exist several limitations for this exp-weighted regression scheme:

- *Optimality-stability trade-off.* As the exponential function $\exp(\cdot)$ grows rapidly, a high-quality action with a large $G(s,a)$ value can lead to an extremely large weight term when $\beta$ is large, causing the explosion of learning loss and destabilizing the training process (illustrated in Figure 1). In practice, many methods mitigate this issue by either using a small $\beta$ or clipping the weighting term (Garg et al., 2023; Xu et al., 2023a; Hansen-Estruch et al., 2023). However, these treatments compromise the optimality of the extracted policy.

- *Inefficient learning.* The training loss becomes dominated by a small number of high-return samples, which is inefficient for policy optimization (Park et al., 2024). Additionally, poor-quality samples still retain positive weight, which can adversely affect policy learning. The constrained optimization objective also makes the learning process highly dependent on the quality of the reference policy $\mu$, thereby limiting the potential for greedy policy optimization.

### 3.2 DICHOTOMOUS DIFFUSION POLICY IMPROVEMENT

To address the drawbacks of the previous weighted regression scheme while preserving its simplicity and scalability, we instead consider a greedified KL-regularized RL objective.

**Greedified policy optimization.** We begin by formulating a greedier learning objective compared to Eq. (2), presented in Eq. (5). At first glance, it appears to be complex; however, as we will show in the later derivation, its resulting closed-form optimal solution can lead to a remarkably elegant form for effective diffusion policy optimization.

$$\max_{\pi} \mathbb{E}_{s\sim d^\pi(s)}\left[\mathbb{E}_{a\sim\pi(a|s)}[G(s,a)] - \frac{1}{\omega\beta}D_{\text{KL}}\left(\pi(\cdot|s)\|\mu(\cdot|s)\cdot\frac{\sigma(\beta G(s,a))}{Z(s)}\right)\right], \tag{5}$$

In this revised objective, we instead regularize policy $\pi$ with a greedified, value-aware reference policy weighted by $\sigma(\beta G(s,a))/Z(s)$, where $Z(s)$ denotes the normalization factor and $\sigma(x) = 1/(1+\exp(-x))$ is the sigmoid function. This design shares a similar spirit with some

offline RL methods that enhance policy performance by regularizing towards a greedier behavior policy or reward-weighted datasets (Singh et al., 2022; Hong et al., 2023; Xu et al., 2025). It is worth noting that we use a bounded and smooth sigmoid function as the weighting function, which greedily assigns high weights to high-return samples while avoiding numerical instability. Moreover, we introduce a new hyperparameter $\omega$, termed the greediness factor, which provides an additional interface for adjusting the greediness of policy extraction. We will reveal its role in the later derivation. Based on the optimization objective in Eq. (5), we can get its closed-form solution as follows:

**Theorem 1.** *The optimal solution for Eq. (5) satisfies:*

$$\pi^\star(a \mid s) \propto \mu(a \mid s) \cdot \sigma\left(\beta G(s, a)\right) \cdot \exp(\omega \cdot \beta G(s, a)). \tag{6}$$

Proof of this theorem can be found in Appendix B. The optimal solution corresponds to a value-aware reference policy with a special weighting scheme, where both $\beta$ and the greediness factor $\omega$ control the level of greediness in the resulting policy. Next, we will show how this solution enables natural decomposition into a pair of dichotomous policies.

**Dichotomous policy extraction.** Leveraging the property of the sigmoid function, it's easy to show:

$$\pi^\star(a \mid s) \propto \mu(a \mid s) \cdot \sigma\left(\beta G(s, a)\right) \cdot \exp(\omega \cdot \beta G(s, a))$$

$$\Leftrightarrow \quad \pi^\star(a \mid s) \propto \mu(a \mid s) \cdot \sigma\left(\beta G(s, a)\right) \cdot \left(\frac{\sigma\left(\beta G(s, a)\right)}{1 - \sigma\left(\beta G(s, a)\right)}\right)^\omega$$

$$\Leftrightarrow \quad \pi^\star(a \mid s) \propto \left[\mu(a \mid s) \cdot \sigma\left(\beta G(s, a)\right)\right]^{1+\omega} / \left[\mu(a \mid s) \cdot (1 - \sigma\left(\beta G(s, a)\right))\right]^\omega. \tag{7}$$

Eq. (7) suggests that the optimal policy can actually be expressed as the ratio of two weighted reference policies with distinct exponents and weighting functions. Specifically, we can define a positive policy $\pi^+$ and a negative policy $\pi^-$ as:

$$\pi^+(a \mid s) \propto \mu(a \mid s) \cdot \sigma\left(\beta G(s, a)\right), \quad \pi^-(a \mid s) \propto \mu(a \mid s) \cdot (1 - \sigma\left(\beta G(s, a)\right)), \tag{8}$$

where the positive policy $\pi^+$ aims to maximize the return and the negative policy $\pi^-$ minimizes it. We call $\pi^+$ and $\pi^-$ *dichotomous policies*, as they share similar form but with opposite focuses. With this definition, the optimal policy can be simply expressed as $\pi^* \propto [\pi^+]^{(1+\omega)} / [\pi^-]^\omega$. Careful readers will notice that both $\pi^+$ and $\pi^-$ are weighted by strictly bounded sigmoid weight functions, instead of the unstable and unbounded exponential weight term $\exp(\beta G(s, a))$ in the optimal solution of the original KL-regularized objective Eq. (3). This means that the decomposed dichotomous policies can be stably trained, precluding loss explosion as discussed in Section 3.1. Moreover, as the positive policy $\pi^+$ prioritizes learning from high-return samples, while the negative policy $\pi^-$ prioritizes learning from low-return samples, we can thus simultaneously utilize both good and bad data for policy optimization, completely resolving the issue of being dominated by high-return samples as in exp-weighted regression, and enabling more efficient learning.

Following Lemma 1, we can train the positive and negative policies $\pi^+$ and $\pi^-$ using two diffusion models with their bounded sigmoid weight functions, parameterized as $\epsilon_{\theta_1}^+$ and $\epsilon_{\theta_2}^-$:

$$\begin{aligned} \mathcal{L}_{\epsilon_{\theta_1}^+} &= \mathbb{E}_{t \sim U[0,1], \epsilon \sim \mathcal{N}(\mathbf{0}, \mathbf{I}), (s,a) \sim \mathcal{D}} \left[\sigma\left(\beta G(s, a)\right) \cdot \left\| \epsilon - \epsilon_{\theta_1}^+(a_t, s, t) \right\|^2\right] \\ \mathcal{L}_{\epsilon_{\theta_2}^-} &= \mathbb{E}_{t \sim U[0,1], \epsilon \sim \mathcal{N}(\mathbf{0}, \mathbf{I}), (s,a) \sim \mathcal{D}} \left[(1 - \sigma\left(\beta G(s, a)\right)) \cdot \left\| \epsilon - \epsilon_{\theta_2}^-(a_t, s, t) \right\|^2\right]. \end{aligned} \tag{9}$$

**Controllable generation.** To sample from the optimal policy $\pi^\star$, note that based on Eqs. (7–8),

$$\log \pi^\star(a \mid s) = (1 + \omega) \log \pi^+(a \mid s) - \omega \log \pi^-(a \mid s) + \log C$$

$$\Rightarrow \quad \nabla_a \log \pi^\star(a \mid s) = (1 + \omega) \nabla_a \log \pi^+(a \mid s) - \omega \nabla_a \log \pi^-(a \mid s), \tag{10}$$

where $C$ is a constant. This shows that the score function of the optimal policy $\pi^\star$ can be expressed as a linear combination of scores of the dichotomous policies, weighted by $\omega$. Due to the inherent connection between the score function and the noise predictor in diffusion model (Ho et al., 2020), we can use $\tilde{\epsilon}(a_t, s, t) = (1 + w)\epsilon_{\theta_1}^+(a_t, s, t) - w\epsilon_{\theta_2}^-(a_t, s, t)$ in the reverse process of diffusion or flow matching for action sampling.

Interestingly, the formulation in Eq. (10) is remarkably similar to classifier-free guidance (CFG) (Ho & Salimans, 2022), a popular method for enhanced conditional diffusion generation, which has the form of $\tilde{\epsilon}(x_t, c, t) = (1 + \omega)\epsilon_\theta(x_t, c, t) - \omega\epsilon_\theta(x_t, t)$, where $\epsilon_\theta(x_t, c, t)$ is a conditioned version of $\epsilon_\theta(x_t, t)$ with conditioning signal $c$. This reveals the inherent connection between our greedified KL-regularized RL objective and the CFG mechanism. Intuitively, our method further strengthens the positive distribution by pushing the negative distribution in the opposite direction, thus enabling flexible control of the optimality level of generated actions with the greediness factor $\omega$ (see illustration in Figure 1). Our final formulation also has some similarity with CFGRL (Frans et al., 2025), which can be perceived as setting $\pi^+ \propto \mu \cdot \mathbb{I}_{A \geq 0}$ and $\pi^- = \mu$ ($A$ is the advantage function). However, their method lacks theoretical backing, and using identical weights for both positive and negative samples limits the greediness of policy optimization, leading to suboptimal performance.

### 3.3 PRACTICAL IMPLEMENTATIONS

**Offline and offline-to-online RL.** For standard multi-step RL settings, we can set $G(s, a)$ as the advantage function $A(s, a)$. In the offline RL setting, the reference policy $\mu$ in Eq. (5) corresponds to the behavior policy $\pi_\beta$ of the offline datasets. In the offline-to-online setting, the reference policy is set as the policy updated in the previous step $\pi_{k-1}$ ($\pi_0$ is the offline pre-trained policy). The algorithm pseudocode and additional implementation details are provided in Appendix C and D.

**End-to-end autonomous driving.** We also implement *DIPOLE* to train a large end-to-end autonomous driving model to demonstrate its scalability to solve real-world complex tasks. Specifically, we employ a non-reactive pseudo-closed-loop simulation based on real-world datasets for policy training. In this setup, the return $G(s, a)$ is defined by a reward function that evaluates trajectory quality based on safety, progress, and comfort. We employ a vision-language model (Florence-2 (Xiao et al., 2024)) as the encoder and a diffusion action head as the decoder (Zheng et al., 2025b). The model processes images from the left-front, front, and right-front cameras, along with language instructions such as "turn left", "turn right", and "go straight". This architecture results in a 1-billion parameter model, which we name *DP-VLA*, and is pre-trained using imitation learning. Subsequently, two separate LoRA modules are applied to the decoder to construct the positive and negative policies, allowing us to leverage Eq. (9) for training. We follow the offline-to-online RL setting to fine-tune the VLA model. Further implementation details are provided in Appendix E.

## 4 EXPERIMENTS

### 4.1 EXPERIMENTS ON RL BENCHMARKS

**Experimental setup.** We evaluate our approach on two commonly-used benchmarks, OGBench (Park et al., 2025a) task suite and ExORL (Yarats et al., 2022) benchmark. OGBench provides challenging robotic locomotion and manipulation tasks, including complex whole-body humanoid control, maze navigation, and object manipulation. We use the default dataset collected by RND (Burda et al., 2019), including tasks in complex high-dimensional state-based domains: Walker, Quadruped, Jaco, and Cheetah. Our evaluation encompasses 30 tasks across 6 domains on OGBench and 9 tasks across 4 domains on ExORL for offline learning, totaling 39 tasks. Finally, we select 4 default tasks across 4 domains on OGBench for offline-to-online validation. Further details are provided in Appendix D.1.

**Baselines.** We use representative baselines across policy types for comprehensive comparison:

- *Gaussian policy.* Standard RL uses Gaussian policies by default. In comparison with standard methods, we select 1) *IQL* (Kostrikov et al., 2022): a typical weighted regression offline RL method. 2) *ReBRAC* (Tarasov et al., 2023): an effective behavior-regularized actor-critic approach incorporates several specific designs tailored for offline learning.

- *Diffusion/Flow policy.* We also include offline RL baselines built on diffusion or flow policies according to the following learning strategies: 1) *IDQL* (Hansen-Estruch et al., 2023) and *IFQL* (Park et al., 2025b): both approaches employ expectile regression for value learning and utilize imitation pre-trained diffusion or flow models with rejection sampling during inference. 2) *FQL* (Park et al., 2025b): a behavior-regularized actor-critic variant that uses flow policy distillation and shows strong performance on OGBench. 3) *CFGRL* (Frans et al., 2025): a recently pro-

Table 1: **ExORL Results.** We report the average score over 8 random seeds. *DIPOLE* achieves the best performance. (w/o rs: without rejection sampling)

| Domain | Task | Gaussian Policy | | Diffusion/Flow Policy | | | | |
|---|---|---|---|---|---|---|---|---|
| | | IQL | ReBRAC | CFGRL | IFQL | FQL | DIPOLE w/o rs | DIPOLE |
| Walker | stand | $603\pm8$ | $461\pm3$ | $782\pm8$ | $873\pm6$ | $801\pm4$ | $793\pm11$ | $953\pm4$ |
| | walk | $444\pm4$ | $208\pm6$ | $608\pm32$ | $844\pm11$ | $755\pm12$ | $679\pm16$ | $910\pm5$ |
| | run | $247\pm10$ | $98\pm2$ | $282\pm6$ | $406\pm8$ | $294\pm11$ | $256\pm12$ | $442\pm9$ |
| Quadruped | walk | $776\pm15$ | $344\pm7$ | $762\pm25$ | $883\pm12$ | $739\pm25$ | $813\pm21$ | $928\pm55$ |
| | run | $485\pm7$ | $344\pm3$ | $571\pm25$ | $595\pm18$ | $503\pm5$ | $560\pm11$ | $657\pm10$ |
| Cheetah | run | $168\pm7$ | $97\pm13$ | $216\pm15$ | $269\pm16$ | $222\pm14$ | $194\pm9$ | $274\pm12$ |
| | run-backward | $146\pm8$ | $85\pm4$ | $262\pm26$ | $310\pm24$ | $231\pm12$ | $227\pm7$ | $350\pm15$ |
| Jaco | reach-top-right | $33\pm2$ | $38\pm13$ | $72\pm6$ | $193\pm9$ | $224\pm17$ | $84\pm5$ | $117\pm18$ |
| | reach-top-left | $30\pm8$ | $59\pm5$ | $46\pm6$ | $181\pm11$ | $222\pm42$ | $63\pm8$ | $110\pm12\alpha$ |

Table 2: **OGBench Results.** We report the aggregate score on all single tasks for each category, averaging over 8 random seeds. DIPOLE achieves best or near-best performance against other baselines across 6 challenging task categories. See appendix D.1 for full results.

| Task Category | Gaussian Policy | | Diffusion/Flow Policy | | | |
|---|---|---|---|---|---|---|
| | IQL | ReBRAC | IDQL | IFQL | FQL | DIPOLE |
| humanoidmaze-medium-navigate (5 tasks) | $33\pm2$ | $2\pm8$ | $1\pm0$ | $60\pm14$ | $58\pm5$ | $68\pm3$ |
| humanoidmaze-large-navigate (5 tasks) | $2\pm1$ | $2\pm1$ | $1\pm0$ | $11\pm2$ | $4\pm2$ | $6\pm2$ |
| antsoccer-arena-navigate (5 tasks) | $8\pm2$ | $0\pm0$ | $12\pm4$ | $33\pm6$ | $60\pm2$ | $57\pm7$ |
| cube-single-play (5 tasks) | $83\pm3$ | $91\pm2$ | $95\pm2$ | $79\pm2$ | $96\pm1$ | $97\pm2$ |
| cube-double-play (5 tasks) | $7\pm1$ | $12\pm1$ | $15\pm6$ | $14\pm3$ | $29\pm2$ | $44\pm7$ |
| scene-play (5 tasks) | $28\pm1$ | $41\pm3$ | $46\pm3$ | $30\pm3$ | $56\pm2$ | $60\pm2$ |

posed policy improvement framework relies on classifier-free guidance, which uses high-quality actions for conditional policy training and unconditional behavior cloning.

For the offline RL setting, we compare our approach with *IQL*, *ReBRAC*, *CFGRL*, *IFQL*, and *FQL* on the ExORL benchmark. We also include a variant, *DIPOLE w/o rs*, which does not use rejection sampling during inference for clear comparison. For *IFQL*, we utilizes the default hyperparameters, and for *FQL*, we select the hyperparameter $\alpha$ reported in previous work (Park et al., 2025b) with optimal performance in ExORL. Additionally, we compare with *IQL*, *ReBRAC*, *IDQL*, *IFQL*, and *FQL* on OGBench to demonstrate our method's effectiveness against state-of-the-art approaches in challenging benchmark.

**Evaluation results.** We evaluate the performance of each method after a fixed number of offline updates for the offline RL setting. Specifically, we report the average return on ExORL and the success rate on OGBench, following the standard evaluation assessment methods (Yarats et al., 2022; Park et al., 2025a). For the offline-to-online evaluation, we assess the final performance on a fixed number of online updates following the formal offline pretraining stage. All evaluations are averaging over 8 random seeds, with $\pm$ indicating standard deviations in tables. We present our results by answering the following questions:

- *Can DIPOLE outperform prior SOTA RL algorithms in offline setting?* Table 1 reports per-task comparison results on ExORL. *DIPOLE* outperforms other baselines in most domains, indicating its capability to fully utilize valuable data in dataset. Specifically, *DIPOLE* fully surpasses *IQL*, indicating its strong improvement over the Gaussian policy-based weighted regression method. Furthermore, *DIPOLE w/o rs* demonstrates better performance compared to *CFGRL*, highlighting the importance of our design for achieving more greedy policy optimization. Finally, Table 2 summarizes the aggregate benchmarking results on OGBench. In most task categories, *DIPOLE* achieves better performance compared to other baselines, demonstrating its strong capability in

Table 3: **OGBench Offline-to-Online Results.** We report the score on the default task for each category, averaging over 8 random seeds. (humanoidmaze-m: humanoidmaze-medium-navigate)

| | Gaussian Policy | | Diffusion/Flow Policy | | |
|---|---|---|---|---|---|
| **Task Category** | IQL | ReBRAC | IFQL | FQL | DIPOLE |
| humanoidmaze-m | $21\pm13 \to 16\pm8$ | $16\pm20 \to 1\pm1$ | $56\pm35 \to 82\pm20$ | $12\pm7 \to 22\pm12$ | $61\pm10 \to 97\pm2$ |
| antsoccer-arena | $2\pm1 \to 0\pm0$ | $0\pm0 \to 0\pm0$ | $26\pm15 \to 39\pm10$ | $28\pm8 \to 86\pm5$ | $43\pm4 \to 90\pm3$ |
| cube-double | $0\pm1 \to 0\pm0$ | $6\pm5 \to 28\pm28$ | $12\pm9 \to 40\pm5$ | $40\pm11 \to 92\pm3$ | $41\pm6 \to 89\pm10$ |
| scene | $14\pm11 \to 10\pm9$ | $55\pm10 \to 100\pm0$ | $0\pm1 \to 60\pm39$ | $82\pm11 \to 100\pm1$ | $97\pm4 \to 100\pm0$ |

Table 4: **NAVSIM Closed-Loop Results**. We scale up *DIPOLE* to a large VLA model, demonstrating its potential for real-world applications. (navtrain/navtest represent different data splits used for trajectory rollout)

| Method | Input | NC↑ | DAC↑ | TTC↑ | Comf.↑ | EP↑ | **PDMS↑** |
|---|---|---|---|---|---|---|---|
| Constant Velocity | - | 68.0 | 57.8 | 50.0 | 100 | 19.4 | 20.6 |
| Ego Status MLP | - | 93.0 | 77.3 | 83.6 | 100 | 62.8 | 65.6 |
| UniAD | Cam | 97.8 | 91.9 | 92.9 | 100.0 | 78.8 | 83.4 |
| PARA-Drive | Cam | 97.9 | 92.4 | 93.0 | 99.8 | 79.3 | 84.0 |
| LFT | Cam | 97.4 | 92.8 | 92.4 | 100 | 79.0 | 83.8 |
| Transfuser | Cam & Lidar | 97.7 | 92.8 | 92.8 | 100.0 | 79.2 | 84.0 |
| Hydra-MDP | Cam & Lidar | 98.3 | 96.0 | 94.6 | 100.0 | 78.7 | 86.5 |
| DP-VLA (ours) | Cam | 98.0 | 97.0 | 94.3 | 100.0 | 82.5 | 88.3 |
| DP-VLA w/ DIPOLE navtrain (ours) | Cam | 98.2 | 98.0 | 95.2 | 100.0 | 83.6 | 89.7 |
| DP-VLA w/ DPPO navtest | Cam | 97.9 | 97.6 | 94.1 | 100.0 | 83.5 | 89.0 |
| DP-VLA w/ DIPOLE navtest (ours) | Cam | 99.2 | 98.7 | 95.6 | 99.8 | 94.2 | 94.8 |

solving challenging long-horizon tasks. These results confirm that weighted regression can effectively achieve greedy policy extraction across robotic locomotion and manipulation RL tasks.

- *How does DIPOLE perform with online finetuning?* Table 3 reports the exact performance variation after 1M of online updates. We demonstrate that our method can be successfully applied to online fine-tuning settings. Compared to *IFQL*, it achieves a higher performance upper bound. When compared to the direct value maximization approach in *FQL*, our method shows competitive performance, demonstrating the effectiveness of our design for achieving both greedy and stable policy optimization. Moreover, we provide pixel-based online fine-tuning results in end-to-end autonomous driving later, further demonstrating the effectiveness of our approach.

Moreover, we refer to Appendix D.4 for ablation studies.

## 4.2 EXPERIMENTS ON AUTONOMOUS DRIVING BENCHMARK

**Experimental setup.** Our method is evaluated on the large-scale real-world autonomous driving benchmark NAVSIM (Dauner et al., 2024) using closed-loop assessment. Following the official evaluation protocol, we report the PDM score (higher indicates better performance), which aggregates five key metrics: *NC* (no-collision rate), *DAC* (drivable area compliance), *TTC* (time-to-collision safety), *Comfort* (acceleration/jerk constraints), and *EP* (ego progress). All methods are tested under the official closed-loop simulator, and results are averaged over the public test split. We also consider an RL application scenario where RL can be applied in human take-over situations or complex environments lacking ground-truth supervision. To address this, we provide a variant of our model trained on the test split without using any ground-truth.

**Baselines.** We select several baselines: 1) *UniAD* (Hu et al., 2023): integrates multiple auxiliary tasks such as tracking, mapping, prediction, and occupancy prediction using transformer blocks, and employs latent representations for planning. 2) *PARA-Drive* (Weng et al., 2024): adopts a parallel architecture design compared to *UniAD*. 3) *Transfuser* (Chitta et al., 2023): fuses image and LiDAR information through a dual-branch architecture and incorporates detection and BEV semantic maps for auxiliary supervision. Its latent variant, *LFT*, replaces LiDAR inputs with learnable embeddings. 4) *Hydra-MDP* (Li et al., 2024): winner of the CVPR2024 Challenge, which uses trajectory anchors and a learned reward model for anchor selection. Moreover, we also consider baselines where the

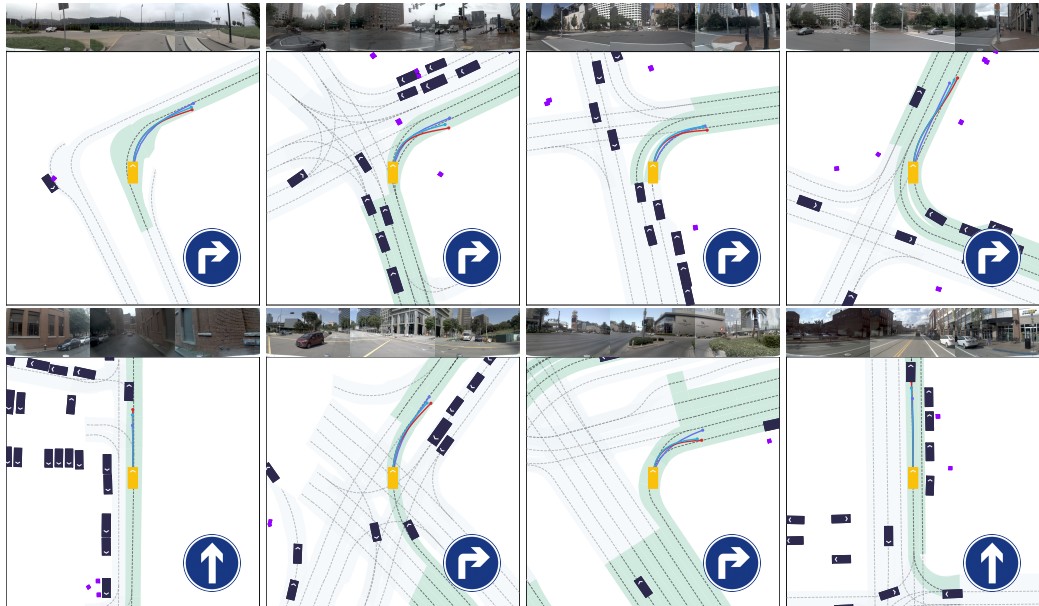

Figure 2: NAVSIM Results: *DP-VLA w/ DIPOLE* fine-tuned model trajectory; ground truth ego trajectory; *DP-VLA* imitation pretrained model trajectory.

agent either maintains its current state or uses a simple MLP for trajectory regression. For our imitation pre-trained VLA model, which directly generates trajectories without post-processing, we denote it as *DP-VLA*. When fine-tuned with DPPO(Ren et al., 2025), we refer to them as *DP-VLA w/ DPPO*. When fine-tuned with our RL algorithm, we refer to it as *DP-VLA w/ DIPOLE*. As mentioned above, we also provide two variants trained on the navtrain and navtest splits.

**Evaluation results.** We present the experimental results in Table 4. Notably, our imitation-based VLA model significantly outperforms other baselines, providing a strong foundation for RL fine-tuning. Building on this, fine-tuning with *DIPOLE* on the navtrain dataset improves the PDMS score by 1.4 points (from 88.3 to 89.7), with gains observed in both safety and progress metrics. Furthermore, *DIPOLE* fine-tuning on navtest scenarios yields a substantial 6.5-point PDMS improvement (from 88.3 to 94.8), demonstrating its potential for real-world autonomous driving applications. These results confirm that even for large-scale policies exceeding 1 billion parameters, *DIPOLE* consistently delivers significant performance improvements through stable and greedy policy optimization. To further illustrate the efficacy of the *DIPOLE* fine-tuned model, we present several cases in Figure 2, where the pretrained model fails but succeeds after *DIPOLE* fine-tuning. Notably, *DIPOLE* enables *DP-VLA* to mitigate compounding errors and low-level controller tracking errors, effectively correcting trajectories to prevent collisions and erratic driving behavior.

## 5 RELATED WORK

Reinforcement fine-tuning of diffusion models remains challenging due to their multi-step diffusion process, primarily in terms of learning stability and computational efficiency. A brute-force solution involves directly optimizing the reward via gradient backpropagation. ReFL (Xu et al., 2023b) optimizes human preference scores for image generation by backpropagating gradients at specific single steps during the reverse process. DRaFT (Clark et al., 2023) extends this approach by applying gradient optimization across multiple steps at the end of the reverse process. Such methods are widely used in motion generation (Karunratanakul et al., 2024), image generation (Prabhudesai et al., 2023), and decision-making tasks (Wang et al., 2022), but suffer from instability due to noisy gradient backpropagation during the denoising process. Some methods avoid gradient computation and instead search for the optimal noise to maximize reward, a strategy referred to as inference-time scaling (Hansen-Estruch et al., 2023; Ma et al., 2025b; Singhal et al., 2025). Recent approaches also utilize RL to directly search for the best noise (Wagenmaker et al., 2025). In both cases, perfor-

mance remains constrained by the capabilities of the pre-trained model. Moreover, DDPO (Black et al., 2024b) treats each noise step as a Gaussian distribution, enabling likelihood estimation and optimization via the REINFORCE (Mohamed et al., 2020) algorithm. DPPO (Ren et al., 2025) optimizes this approach and extends it to multi-step MDPs using PPO (Schulman et al., 2017) for policy improvement. These methods rely on Gaussian approximations that require sufficiently small sampling steps, resulting in inefficient training. Some methods (Lee et al., 2023; Kang et al., 2023; Zheng et al., 2024; Ma et al., 2025a; Zheng et al., 2025a) use KL-regularized RL (Kostrikov et al., 2022; Peng et al., 2019), whose solution leads to a simple weighted regression loss. However, these approaches often face a trade-off between greediness and stability.

## 6 CONCLUSION

We propose *DIPOLE*, an RL method that enables stable and controllable diffusion policy optimization. We revisit KL-regularized RL, which suffers from a trade-off between greediness and stability, and introduce a greedified policy regularization scheme. This scheme decomposes the optimal policy into dichotomous policies with stable training losses. During inference, actions are generated by linearly combining the scores of these policies, enabling controllable greediness. We evaluate *DIPOLE* on widely used RL benchmarks to demonstrate its effectiveness and also train a large VLA model for end-to-end autonomous driving, highlighting its potential for real-world applications. Due to space limit, more discussion on limitations and future direction can be found in Appendix F.

## ACKNOWLEDGEMENT

This work is supported by Xiaomi EV and National Natural Science Foundation of China under Grant No.62276260, funding from Wuxi Research Institute of Applied Technologies, Tsinghua University under Grant 20242001120 and the Xiongan AI Institute. Furthermore, we would like to thank Jianwei Cui and Kun Ma in Xiaomi EV for their resource support. We would like to express our gratitude to Bin Huang, Enguang Liu and Jianlin Zhang in Xiaomi EV for their valuable discussion.

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

## A  LLM USAGE

In this paper, we employed Large Language Models (LLMs) solely for polishing the writing. No parts of the technical content, experimental results, or conclusions were generated by LLMs.

## B  THEORETICAL INTERPRETATIONS

We define our problem under the reinforcement learning problem presented as a Markov Decision Process (MDP) (Sutton et al., 1998) given by $\mathcal{M} = (\mathcal{S}, \mathcal{A}, \mathcal{P}, r, \gamma)$, which comprises a state space $\mathcal{S}$, an action space $\mathcal{A}$, a state transition $\mathcal{S}$, a reward function $r$ and a discount factor $\gamma$. In this setting, a policy is a probability distribution of actions conditioned on a state. In addition, we assume that all policies induce an irreducible Markov Chain, with any two states reachable from each other by a sequence of transitions that have positive probability. Our goal is to find a policy $\pi$ that maximizes a predefined action evaluation criteria $G$, constrained on a reference policy.

**Theorem 1.** *The optimal solution for Eq. (5) satisfies:*

$$\pi^\star(a \mid s) \propto \mu(a \mid s) \cdot \sigma\left(\beta G(s,a)\right) \cdot \exp(\omega \cdot \beta G(s,a)). \tag{6}$$

*Proof.* Consider the optimization problem Eq. (5) with constraints on the probability distribution:

$$\max_{\pi} \ \mathbb{E}_{s \sim d^\pi(s)}\left[\mathbb{E}_{a \sim \pi(a|s)}[G(s,a)] - \tfrac{1}{\omega\beta}\, D_{\mathrm{KL}}\big(\pi(\cdot \mid s) \,\|\, \mu(\cdot \mid s)\tfrac{\sigma(\beta G(s,a))}{Z(s)}\big)\right]$$

$$\text{s.t.} \quad \int_a \pi(a \mid s)\, da = 1, \ \ \forall s \tag{11}$$

$$\pi(a \mid s) \geq 0, \ \ \forall s, a$$

The Lagrangian is given by:

$$\begin{aligned}
\mathcal{L}(\pi, \alpha_s, \gamma_{s,a}) &= \int_s d^\pi(s) \int_a \pi(a \mid s) G(s,a)\,\mathrm{d}a\,\mathrm{d}s \\
&\quad - \int_s d^\pi(s)\left[\frac{1}{\omega\beta}\int_a \pi(a \mid s) \log\left(\frac{\pi(a \mid s)Z(s)}{\mu(a \mid s)\sigma(\beta G(s,a))}\right)\mathrm{d}a\right]\mathrm{d}s \\
&\quad + \int_s \alpha_s\left(\int_a \pi(a \mid s)\mathrm{d}a - 1\right)\mathrm{d}s + \int_{s,a} \gamma_{s,a}\pi(a \mid s)\mathrm{d}a\mathrm{d}s
\end{aligned} \tag{12}$$

Take the derivative over $\pi(a \mid s)$ and set to zero:

$$\begin{aligned}
\frac{\partial \mathcal{L}}{\partial \pi(a \mid s)} &= G(s,a) - \frac{1}{\omega\beta}\left(\log \pi(a \mid s) + 1 - \log\frac{\mu(a \mid s)\sigma(\beta G(s,a))}{Z(s)}\right) + \alpha_s + \gamma_{s,a} \\
&= 0
\end{aligned} \tag{13}$$

Solve the equation and one can obtain the optimal policy as:

$$\pi^\star(a \mid s) = \mu(a \mid s)\sigma(\beta G(s,a))\exp(\omega\beta G(s,a)) \cdot \exp\left(\omega\beta\frac{\alpha_s + \gamma_{s,a}}{d^\pi(s)} - 1 - \log Z(s)\right) \tag{14}$$

Note that since we assume all policies induce irreducible Markov chain, thus $d^\pi(s) > 0, \ \forall s$. Consider the support of $\mu$ with positive probability and the final resulted optimal policy satisfies:

$$\pi^\star(a \mid s) \propto \mu(a \mid s) \cdot \sigma(\beta G(s,a)) \cdot \exp(\omega \cdot \beta G(s,a)) \tag{15}$$

$\square$

As shown in Eq. (10), the score function of the target optimal policy and the dichotomous policies satisfy the linear combination at $t = 0$:

$$\epsilon_0^\star(a|s) = (1+\omega)\epsilon_0^+(a|s) - \omega\epsilon_0^-(a|s) \tag{16}$$

With the marginal condition satisfied, one can obtain the exact intermediate score by (Zheng & Lan, 2024):

$$\epsilon_t^\star(a_t|s) = (1+\omega)\epsilon_t^+(a_t + \omega\Delta a_t|s) - \omega\epsilon_t^-(a_t + (1+\omega)\Delta a_t|s) \tag{17}$$

where $\Delta a_t$ is a non-linear correction term satisfying the following equation:

$$\Delta a_t = \sqrt{1 - e^{-t}}(\epsilon_t^-(a_t + (1+\omega)\Delta a_t|s) - \epsilon_t^+(a_t + \omega\Delta a_t|s)) \tag{18}$$

However, solving this non-linear equation requires iteration on the learned neural networks, which is expensive. In practice, one can directly sample using the direct combination of positive and negative score functions, as an approximation when $\omega$ is relatively small (Ho & Salimans, 2022):

$$\epsilon_t^\star(a|s) \approx (1+\omega)\epsilon_t^+(a|s) - \omega\epsilon_t^-(a|s) \tag{19}$$

## C  ALGORITHM PSEUDOCODE

---
**Algorithm 1** Training

---
**while** not converged **do**
  Collect data, or use offline data $\mathcal{D}$.
  $\epsilon \sim \mathcal{N}(\mathbf{0}, \mathbf{I})$, $t \sim U[0,1]$
  $a_t \leftarrow$ diffusion/flow forward process
  $\theta_1 \leftarrow \theta_1 - \lambda\nabla_{\theta_1}\left[\sigma(\beta G) \cdot \left\|\epsilon - \epsilon_{\theta_1}^+(a_t, s, t)\right\|^2\right]$
  $\theta_2 \leftarrow \theta_2 - \lambda\nabla_{\theta_2}\left[(1 - \sigma(\beta G)) \cdot \left\|\epsilon - \epsilon_{\theta_2}^-(a_t, s, t)\right\|^2\right]$
**end while**

---

---
**Algorithm 2** Sampling

---
$a_1 \sim \mathcal{N}(\mathbf{0}, \mathbf{I})$
$t \leftarrow 1$
**for** $n \in [1, \dots, N]$ **do**
  $\tilde{\epsilon} = (1+w)\epsilon_{\theta_1}^+(a_t, s, t) - w\epsilon_{\theta_2}^-(a_t, s, t)$
  $t \leftarrow t - (n/N)$
  $a_t \leftarrow$ diffusion/flow reverse process, given $\tilde{\epsilon}$
**end for**
**return** $a_0$

---

## D  DETAILS ON RL BENCHMARKS

### D.1  EXPERIMENTAL DETAILS

In this section, we provide the experimental details, including benchmarks, datasets, and tasks. Our experiments span two primary benchmarks: ExORL (Yarats et al., 2022) and OGBench (Park et al., 2025a)

**ExORL.** ExORL consists of datasets collected by multiple unsupervised RL agents (Laskin et al., 2021) on the DeepMind Control Suite (Tassa et al., 2018). We utilize datasets collected by unsupervised RL algorithms *RND* (Burda et al., 2019) across four domains (*Walker*, *Jaco*, *Quadruped*, and *Cheetah*). For each environment, we use the full dataset with all transitions from each dataset.

- **Walker** (locomotion): A bipedal robot with 24-dimensional states (joint positions/velocities) and 6-dimensional actions. Test tasks include *run*, *stand*, and *walk*. Rewards combine dense objectives: maintaining torso height (*stand*) and achieving target velocities (*Run/Walk*).

- **Quadruped** (locomotion): A four-legged robot with 78-dimensional states and 12-dimensional actions. Tasks include *run* and *walk*, with rewards for torso stability and velocity tracking.

- **Jaco** (goal-reaching): A 6-DoF robotic arm with 55-dimensional states and 6-dimensional actions. Tasks involve reaching four target positions (*Top Left/Right*) using sparse rewards based on proximity to goals.

- **Cheetah** (locomotion): A running planar biped with 17-dimensional states consisting of positions and velocities of robot joints, and 6-dimensional actions. The reward is linearly proportional to the forward velocity. We consider tasks *run* and *run backward* for evaluation.

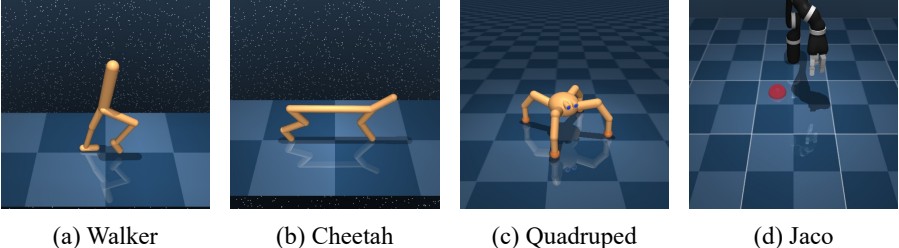

(a) Walker      (b) Cheetah      (c) Quadruped      (d) Jaco

Figure 3: **ExORL environments.** We experiment on 4 high-dimensional complex domains: Walker, Cheetah, Quadruped, and Jaco Arm.

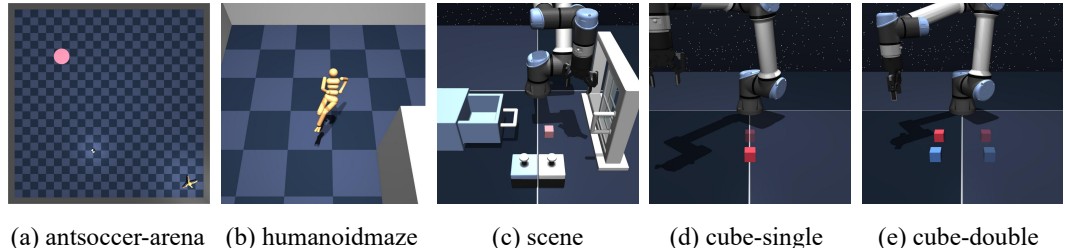

(a) antsoccer-arena    (b) humanoidmaze    (c) scene    (d) cube-single    (e) cube-double

Figure 4: **OGBench environments.** We experiment on 5 complex domains: antsoccer-arena, humanoidmaze, scene, cube-single, and cube-double.

**OGBench.** OGBench is designed for offline goal-conditioned RL, containing multiple challenging tasks across robotic manipulation, navigation, and locomotion. We use 30 state-based manipulation and navigation tasks from 6 domains (humanoidmaze-medium-navigate, humanoidmaze-large-navigate, cube-single-play, cube-double-play, scene-play, and antsoccer-arena-navigate). Each domain contains 5 different tasks, and one is set as the default task. To be compatible with standard offline RL settings, we leverage its single-task variant. We evaluate offline performance on all single tasks and online fine-tuning performance on the default tasks of the selected 4 challenging domains.

- **humanoidmaze** (navigation):.Controlling a 21-DoF Humanoid agent to reach a goal position in a given maze.
- **cube-play** (manipulation): Controlling a robot arm to pick and place cube-shaped blocks in order to assemble designated target configurations.
- **scene-play** (manipulation): Long-horizon control of multiple objects, including cube block, a window, a drawer, and two button locks.
- **antsoccer-arena** (navigation): Controlling an Ant agent to dribble a soccer ball. The agent must also carefully control the ball while navigating the environment.

### D.2 IMPLEMENTATION DETAILS

We implement DIPOLE in JAX on top of FQL (Park et al., 2025b) and CFGRL (Frans et al., 2025).

**Architectures.** We train 5 neural networks in parallel: two policy networks (positive policy and negative policy)value networks (two Q-value estimators and one V-value estimator). We use three-layer multi-layer perceptron (MLP) with 512 hidden dimensions for both the policy networks and the value networks. We select the flow policy for the evaluation of our approach due to its efficient training process. Our flow policy is based on linear paths and uniform time sampling.

**Value Learning.** In all experiments, we learn an optimal $V$-value by IQL-style expectile regression. For $Q$-value learning method, we compute the target by optimized $V$-value in ExORL, and the the traditional temporal difference (TD) target Q in OGBench separately. Full details of hyperparameter settings are provided in Table 5.

**Policy extraction and action reweighting.** In RL setting, one of the critical selections of $G(s, a)$ in Eqs. (7) is the advantageous function, i.e. $A(s, a) = Q(s, a) - V(s)$. To induce a more flexible

and controllable learning process, we additionally introduce a tunable hyperparameter to shift the distribution of $G(s, a)$. Specifically, the weighting function becomes $\sigma(\beta G(s, a) + k)$ for positive policy and $1 - \sigma(\beta G(s, a) + k)$ for negative policy. The full details of per-task hyperparameters are provided in Table 6.

We implement rejection sampling for inference time policy output. Specifically, we sample $N$ actions for a single state input and select the action that has the highest Q-value:

$$a^\star \triangleq \arg \max_{a \in \{a^{(1)}, \ldots, a^{(N)} \sim \pi(s)\}} Q(s, a). \tag{20}$$

**Evaluation.** We report the average return on ExORL and the success rate on OGBench, following the standard evaluation assessment methods (Yarats et al., 2022; Park et al., 2025a). For offline RL performance evaluation, we fix the gradient to be 1M and report the final score. For the offline-to-online RL performance evaluation, we report both the 1M offline score and the 1M online score.

**Computation resource.** We train our model on NVIDIA A6000 GPUs. Training a single task on one GPU takes approximately 0.5 hours on ExORL and 1.5 hours on OGBench.

## D.3 HYPERPARAMETERS

In this section, we provide the detailed hyperparameter setup in Table 5 and Table 6. In our experiments, the model architecture and basic algorithm hyperparameters remain unchanged, as detailed in Table 5. To encourage a better trade-off between greediness and stability, we adopt domain-specific hyperparameters, including expectile parameters $\tau$, beta $\beta$, shift factor $k$, and discount factor $\gamma$, as detailed in Table 6.

Table 5: General hyperparameters used for DIPOLE

|  | Hyperparameter | Value |
|---|---|---|
| | Optimizer | Adam |
| | Policy learning rate | 3e-4 |
| | Value learning rate | 3e-4 |
| | Offline learning steps | 1,000,000 |
| DIPOLE Hyperparameters | Online fintuning steps | 0 (offline), 1,000,000 (offline-to-online) |
| | Mini-batch | 512 (ExORL), 256 (OGBench) |
| | Soft update factor $\lambda$ | 0.005 |
| | Diffusion/Flow steps $T$ | 32 (ExORL), 10 (OGBench) |
| | Clip Q | false (cube-single-play; scene-play), true (others) |
| | Policy MLP hidden dimension | [512, 512, 512] |
| Architecture | Value MLP hidden dimension | [512, 512, 512] |
| | Activation function | tanh |

## D.4 ABLATION STUDY

**Hyperparameter.** Both hyperparameters beta $\beta$, shift factor $k$, expectile factor $\tau$, and rejection sampling action number $N$ are important for DIPOLE's performance. In Figure 5, we present the performance changes when fixing single action sampling, and present the performance changes for the default action sampling number when tuning the expectile factor. We further ablate the influence of beta $\beta$ and CFG scale $\omega$ on DIPOLE. We conducted experiments on ExORL benchmark, average over 3 random seeds. Both the impact of $\beta$ and $\omega$ are within a similar pattern, which shows a easy-tuning property of DIPOLE.

## D.5 ADDITIONAL RESULTS

**OGBench full results.** In this section, we provide the full experimental results for all single tasks on the OGBench, as shown in Figure 7. All results are averaged over 8 random seeds. We report the mean and standard deviation of the final score, after 1M gradient steps.

Table 6: Task-specific hyperparameters for DIPOLE.

| Task Category | beta $\beta$ | shift factor $k$ | discount $\gamma$ | expectile $\tau$ | sample actions $N$ |
|---|---|---|---|---|---|
| OGBench-humanoidmaze-medium-navigate | 1 | 0 | 0.99 | 0.9 | 4 |
| OGBench-humanoidmaze-large-navigate | 1 | 0 | 0.995 | 0.9 | 8 |
| OGBench-antsoccer-arena-navigate | 1 | 0 | 0.995 | 0.9 | 4 |
| OGBench-cube-single-play | 0.5 | 1 | 0.99 | 0.9 | 2 |
| OGBench-cube-double-play | 0.5 | 0.5 | 0.99 | 0.9 | 2 |
| OGBench-scene-play | 1 | 1 | 0.99 | 0.9 | 2 |
| ExORL-walker-walk | 3.5 | -2 | 0.99 | 0.95 | 32 |
| ExORL-walker-stand | 4.5 | -2 | 0.99 | 0.99 | 32 |
| ExORL-walker-run | 4.5 | -2 | 0.99 | 0.9 | 32 |
| ExORL-quadruped-walk | 3 | 0 | 0.99 | 0.9 | 32 |
| ExORL-quadruped-run | 4 | -2 | 0.99 | 0.9 | 32 |
| ExORL-jaco-reach-top-left | 1 | 0 | 0.99 | 0.9 | 32 |
| ExORL-jaco-reach-top-right | 1 | -1 | 0.99 | 0.9 | 32 |
| ExORL-cheetah-run | 4 | -1 | 0.99 | 0.9 | 32 |
| ExORL-quadruped-run-backward | 4 | -1 | 0.99 | 0.9 | 32 |

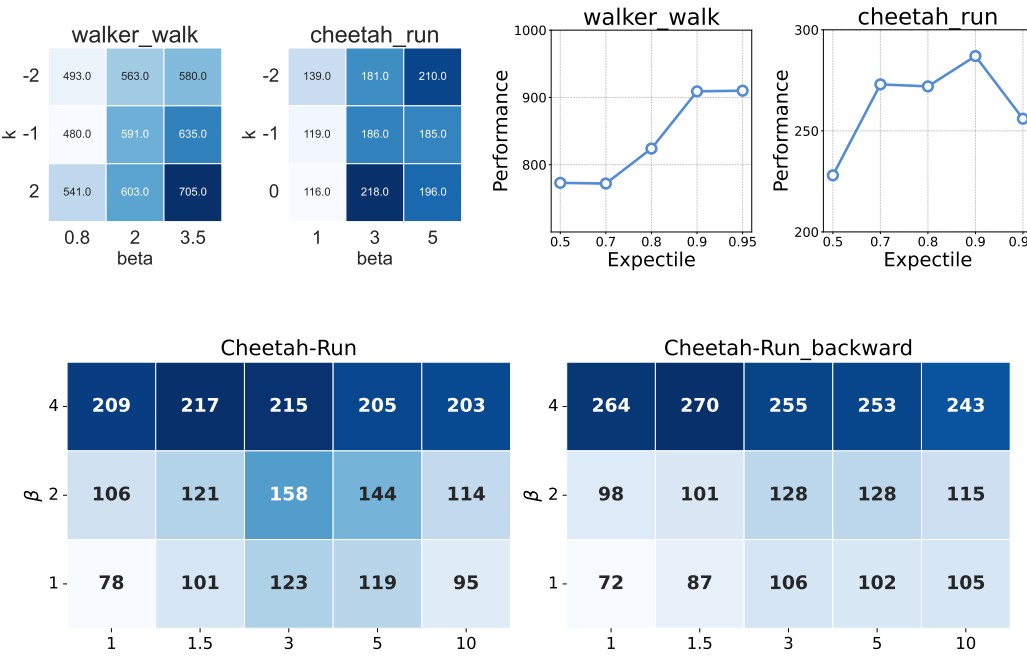

Figure 5: Top-left: ablation on $\beta$ and $k$; Top-right: ablation on expectile $\tau$; Bottom: ablation on $w$ and $\beta$

**OGBench offline-to-online learning curves.** We also present the training curves of *DIPOLE*, including both the 1M offline gradient steps and 1M online gradient steps, as shown in Figure 6.

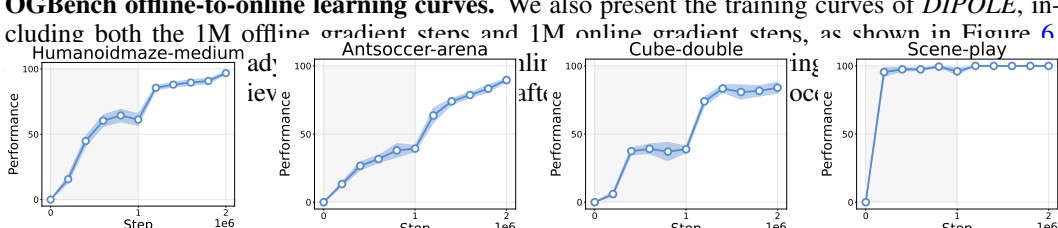

Figure 6: **Offline-to-online visualization.** DIPOLE presents a stable fine-tuning process on OGBench.

Table 7: OGBench full results.

| Task Category | Gaussian Policy | | Diffusion/Flow Policy | | | |
|---|---|---|---|---|---|---|
| | IQL | ReBRAC | IDQL | IFQL | FQL | DIPOLE |
| humanoidmaze-medium-navigate-singletask-task1 | 32±7 | 16±9 | 1±1 | 69±19 | 19±12 | 63±6 |
| humanoidmaze-medium-navigate-singletask-task2 | 41±9 | 18±16 | 1±1 | 85±11 | 94±3 | 91±2 |
| humanoidmaze-medium-navigate-singletask-task3 | 25±5 | 36±13 | 0±1 | 49±49 | 74±18 | 88±4 |
| humanoidmaze-medium-navigate-singletask-task4 | 0±1 | 15±16 | 1±1 | 1±1 | 3±4 | 1±1 |
| humanoidmaze-medium-navigate-singletask-task5 | 66±4 | 24±20 | 1±1 | 98±2 | 97±2 | 96±2 |
| humanoidmaze-large-navigate-singletask-task1 | 3±1 | 2±1 | 0±0 | 6±2 | 7±6 | 20±5 |
| humanoidmaze-large-navigate-singletask-task2 | 0±0 | 0±0 | 0±0 | 0±0 | 0±0 | 0±0 |
| humanoidmaze-large-navigate-singletask-task3 | 7±3 | 8±4 | 3±1 | 48±10 | 11±7 | 7±3 |
| humanoidmaze-large-navigate-singletask-task4 | 1±0 | 1±1 | 0±0 | 1±1 | 2±3 | 1±1 |
| humanoidmaze-large-navigate-singletask-task5 | 1±1 | 2±2 | 0±0 | 0±0 | 1±3 | 2±4 |
| antsoccer-arena-navigate-singletask-task1 | 14±5 | 0±0 | 44±12 | 61±25 | 77±4 | 82±7 |
| antsoccer-arena-navigate-singletask-task2 | 17±7 | 0±1 | 15±12 | 75±5 | 88±3 | 74±5 |
| antsoccer-arena-navigate-singletask-task3 | 6±4 | 0±0 | 0±0 | 14±22 | 61±6 | 55±8 |
| antsoccer-arena-navigate-singletask-task4 | 3±2 | 0±0 | 0±1 | 16±9 | 39±6 | 40±10 |
| antsoccer-arena-navigate-singletask-task5 | 2±2 | 0±0 | 0±0 | 0±1 | 36±9 | 32±5 |
| cube-single-play-singletask-task1 | 88±3 | 89±5 | 95±2 | 79±4 | 97±2 | 97±2 |
| cube-single-play-singletask-task2 | 85±8 | 92±4 | 96±2 | 73±3 | 97±2 | 98±2 |
| cube-single-play-singletask-task3 | 91±5 | 93±3 | 99±1 | 88±4 | 98±2 | 99±2 |
| cube-single-play-singletask-task4 | 73±6 | 92±3 | 93±4 | 79±6 | 94±3 | 94±5 |
| cube-single-play-singletask-task5 | 78±9 | 87±8 | 90±6 | 77±7 | 93±3 | 96±3 |
| cube-double-play-singletask-task1 | 27±5 | 45±6 | 39±19 | 35±9 | 61±9 | 68±7 |
| cube-double-play-singletask-task2 | 1±1 | 7±3 | 16±10 | 9±5 | 36±6 | 44±10 |
| cube-double-play-singletask-task3 | 0±0 | 4±1 | 17±8 | 8±5 | 22±5 | 51±6 |
| cube-double-play-singletask-task4 | 0±0 | 1±1 | 0±1 | 1±1 | 5±2 | 6±2 |
| cube-double-play-singletask-task5 | 4±3 | 4±2 | 1±1 | 17±6 | 19±10 | 50±8 |
| scene-play-singletask-task1 | 94±3 | 95±2 | 100±0 | 98±3 | 100±0 | 100±0 |
| scene-play-singletask-task2 | 12±3 | 50±13 | 33±14 | 0±0 | 76±9 | 96±3 |
| scene-play-singletask-task3 | 32±7 | 55±16 | 94±4 | 54±19 | 98±1 | 99±1 |
| scene-play-singletask-task4 | 0±1 | 3±3 | 4±3 | 0±0 | 5±4 | 5±6 |
| scene-play-singletask-task5 | 0±0 | 0±0 | 0±0 | 0±0 | 0±0 | 0±1 |

# E  DETAILS ON E2E AD BENCHMARKS

## E.1  MODEL ARCHITECTURE

We employ the pretrained Florence-2-large model (Xiao et al., 2024) as the visual-language encoder, paired with a 475M-parameter Diffusion Transformer as the action decoder. The visual input comprises images from Front, Front-Left, and Front-Right perspectives, while the language input consists of driving commands provided by the dataset. Encoder output tokens are processed by the action decoder through a cross-attention block, which ultimately generates the predicted trajectory.

## E.2  TRAINING PROCEDURE

The training process consists of two phases: pretraining and reinforcement learning fine-tuning. In the pretraining phase, we utilize trainval frames from the NAVSIM dataset to jointly train the encoder and decoder using a diffusion loss objective. During the RL fine-tuning phase, the encoder is frozen, and two Low-Rank Adaptation (LoRA) adapters—a positive adapter and a negative adapter—are incorporated into every linear projection of the attention and MLPs. The NavSim benchmark's PDMS score serves as the direct optimization target. Every 10 epochs, the replay buffer is cleared, and new model rollout trajectories and corresponding rewards are collected based on one epoch of data samples. For each data sample, the model generates $g$ trajectories, which are used to train the LoRA adapters over the subsequent 9 epochs. For each trajectory, we obtain its PDMS score vector $\mathbf{r} = \{r_1, r_2, \ldots, r_g\}$, enabling estimation of the advantage function (Shao et al., 2024):

$$G(s, a) = A(s, a) = \frac{r_i - mean(\mathbf{r})}{std(\mathbf{r})} \tag{21}$$

### E.3 INFERENCE AND EVALUATION

During inference, the *DP-VLA* encoder extracts features from the input images and driving commands. The denoising process is solved using the DPM-Solver(Lu et al., 2022), with the action decoder iteratively predicting the denoised trajectory over 10 steps to produce the final clean trajectory. For evaluation on the NAVSIM benchmark, the proposed trajectory is fed into an LQR tracker and dynamics model to compute the posterior trajectory. The final PDM score is derived from this posterior trajectory, satisfying the benchmark's evaluation criteria (Dauner et al., 2024).

### E.4 HYPERPARAMETERS

We summarize the training details of DP-VLA in Table 8. LoRA adapters are applied to all linear projections, including the QKVO projections in Attentions and Linears in FFNs.

Table 8: DIPOLE hyperparameters used in DP-VLA

|  | Hyperparameter | Value |
|---|---|---|
| Pre-train Hyperparameters | Optimizer | AdamW |
| | Learning rate | 1e-4 |
| | Learning epochs | 100 |
| | Mini-batch | 16 |
| DIPOLE navtrain Hyperparameters | Optimizer | AdamW |
| | Learning rate | 1e-4 |
| | Learning steps | 2.069k |
| | Mini-batch | 56 |
| | Group size $g$ | 10 |
| DIPOLE navtest Hyperparameters | Optimizer | AdamW |
| | Learning rate | 1e-4 |
| | Learning steps | 11.52k |
| | Mini-batch | 4 |
| | Group size $g$ | 25 |
| LoRA Hyperparameters | Rank | 16 |
| | Alpha | 16 |
| | Dropout | 0.0 |
| | Num. params(Each/Total) | 6.68M/13.37M |
| | Ratio params(Each/Total) | 1.4%/2.8% |

### E.5 MORE CASES

We selected representative scenarios from navTest and superimposed the trajectories of DP-VLA before and after DIPOLE fine-tuning to visualize the behavioral differences in Figure 7. It can be observed that while the pre-fine-tuned model exhibits off-road driving or collisions, these rule-violating and dangerous behaviors are effectively rectified after fine-tuning.

## F LIMITATION & DISCUSSION & FUTURE WORK

Here, we discuss the limitations, potential solutions, and promising future directions of our work. While this paper presents a policy optimization method for achieving both greedy and stable policy training, we observe that performance is highly dependent on the quality of the value function. Developing effective value function estimation methods for diffusion-based reinforcement learning remains an important direction for future research. Additionally, our approach remains within the behavior-regularized optimization framework, which inherently constrains the policy relative to a reference policy. We attempt to address this limitation through controllable greediness and a greedified optimization objective. While our method has demonstrated applicability in complex autonomous driving tasks, we anticipate its potential extension to other domains.

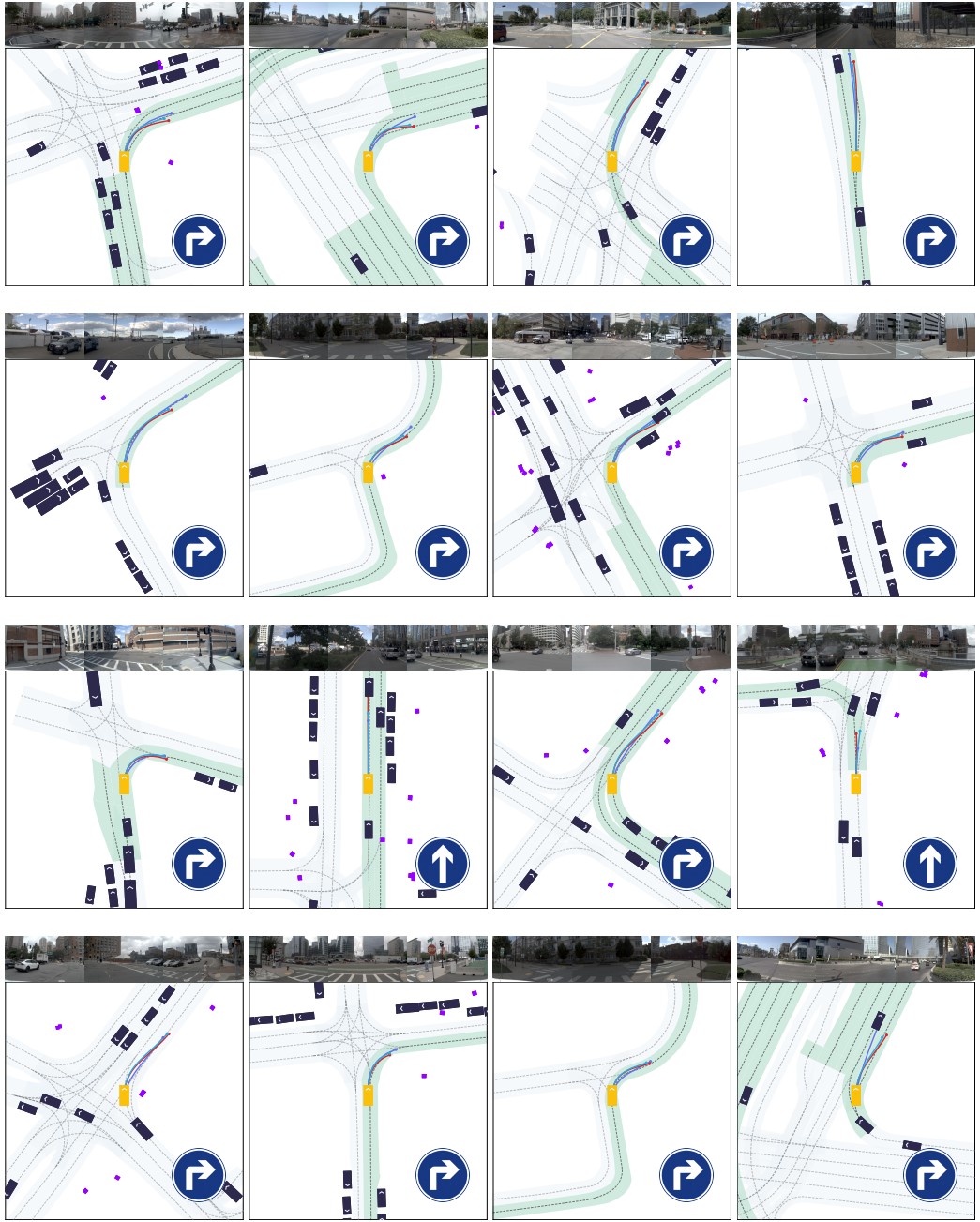

Figure 7: NAVSIM Results: *DP-VLA w/ DIPOLE* fine-tuned model trajectory; ground truth ego trajectory; *DP-VLA* imitation pretrained model trajectory.

