# OpenReview forum: "Dichotomous Diffusion Policy Optimization"
_ICLR.cc/2026/Conference — ICLR 2026 Poster_

### Official Review · Reviewer_ZjGt · 2025-10-30

**Soundness:** 2
**Presentation:** 3
**Contribution:** 2
**Rating:** 4
**Confidence:** 4

**Summary:**

This paper proposes DIPOLE (Dichotomous diffusion Policy improvement), a novel reinforcement learning algorithm for training diffusion-based policies. The authors identify a key challenge in prior work: directly optimizing the standard KL-regularized RL objective,
$$\max_{\pi} \mathbb{E}[G(s,a)] - \frac{1}{\beta} D_{KL}(\pi||\mu)$$
is difficult because its closed-form solution, $\pi^*(a|s) \propto \mu(a|s) \cdot \exp(\beta G(s,a))$, leads to an unstable weighted regression loss where the $\exp(\cdot)$ term can explode.
To overcome this, DIPOLE proposes a new "greedified" KL-regularized objective (Eq. 5). The key insight is that the optimal solution to this new objective can be decomposed into a ratio of two "dichotomous" policies:
1. A "positive" policy $\pi^+ \propto \mu(a|s) \cdot \sigma(\beta G(s,a))$
2. A "negative" policy $\pi^- \propto \mu(a|s) \cdot (1 - \sigma(\beta G(s,a)))$

Critically, the authors claim these policies can be trained stably using diffusion models with bounded sigmoid weights ($\sigma$ and $1-\sigma$), solving the stability-optimality trade-off.
Furthermore, the paper shows that sampling from the optimal policy $\pi^*$ can be achieved by linearly combining the scores of the two dichotomous policies, resulting in an inference rule analogous to Classifier-Free Guidance (CFG):
$$\tilde{\epsilon}(a_t, s, t) = (1 + \omega) \epsilon_{\theta_1}^{+}(a_t, s, t) - \omega \epsilon_{\theta_2}^{-}(a_t, s, t)$$
The authors demonstrate DIPOLE's effectiveness on ExORL and OGBench benchmarks and scale it successfully to a 1-billion parameter vision-language-action (VLA) model for autonomous driving.

**Strengths:**

1. **Originality**: The primary strength is the novel formulation. The idea of decomposing the optimization into a reward-maximizing ($\pi^+$) and a reward-minimizing ($\pi^-$) policy is creative and provides a new lens for policy optimization.
2. **Strong Empirical Results & Scalability**: The method clearly works well in practice. The successful application of DIPOLE (using LoRA) to a 1-billion parameter VLA model is a significant achievement and demonstrates the method's practical utility for fine-tuning large models with RL.

**Weaknesses:**

1. **Questionable Necessity (Unsound Premise)**: The paper's motivation is that the standard KL-regularized objective (Eq. 2) is unusable because its solution (Eq. 3) implies an unstable weighted regression (Eq. 4). This implicitly assumes that weighted regression is the only way to optimize this objective. This premise is challenged by recent work like BDPO (Gao et al., 2025, ICML), which tackles the exact same standard objective. BDPO shows that by decomposing the $D_{KL}$ term along the diffusion path ($D_{KL}[p_{0:N}^\pi || p_{0:N}^\nu]$), the objective becomes a sum of per-step, analytic KL divergences $\sum_n D_{KL}[p_{n-1|n}^\pi || p_{n-1|n}^\nu]$. Since these per-step transitions are Gaussian, this penalty becomes a simple, stable, analytic MSE between the predicted noise vectors (Eq. 17 in BDPO). This suggests that DIPOLE's "greedified" objective (Eq. 5) is an overly complex solution to a problem that has a simpler, more direct solution within the original, standard RL framework.
2. **Flawed Training Mechanism (Sigmoid Saturation)**: The core of DIPOLE's solution is the replacement of $\exp(\beta G)$ with the bounded $\sigma(\beta G)$ (Eq. 9). This introduces a new, critical problem: signal saturation. The sigmoid function saturates, meaning its output approaches 1 for all values above a certain threshold (e.g., $\sigma(10) \approx \sigma(20) \approx 1.0$). This means the training loss for the positive policy $\epsilon^+$ loses all gradient information that distinguishes "good" actions from "excellent" actions. The network is not learning a fine-grained reward landscape, but rather a near-binary classification of "good" (weight $\approx 1$) vs. "bad" (weight $\approx 0$).
This saturation flaw directly contradicts the goal of the inference step (Eq. 10). The inference mechanism $\tilde{\epsilon} = (1+\omega)\epsilon^+ - \omega\epsilon^-$ relies on the "greediness factor" $\omega$ to amplify the difference between the two policies. However, if $\epsilon^+$ has already lost the high-reward gradient information due to saturation, $\omega$ is merely amplifying a "blurry" or "clipped" signal. It's unclear how this can steer the policy towards truly optimal actions if the network was never trained to distinguish them in the first place.
3. **Gap Between Theory and Practice (The $k$ parameter)**: This saturation flaw is strongly corroborated by the implementation details. The paper's theoretical derivation (Eq. 9) relies purely on $\sigma(\beta G)$. However, Appendix D.2 reveals the practical use of a modified weight, $\sigma(\beta G + k)$. This 'shift factor' $k$, which is absent from the main theory, serves as strong evidence that the sigmoid-based weighting is not robust. It implicitly confirms the saturation problem, as the model's performance is highly sensitive to the distribution of $G(s,a)$. The mechanism is therefore not as 'principled' as claimed, requiring an ad-hoc hyperparameter to manually shift the sigmoid's non-saturating region to align with the data.

**References**

Chen-Xiao Gao, Chenyang Wu, Mingjun Cao, Chenjun Xiao, Yang Yu, Zongzhang Zhang Proceedings of the 42nd International Conference on Machine Learning, PMLR 267:18630-18657, 2025.

**Questions:**

1. On the Premise (re: BDPO): The paper's motivation rests on the instability of the $\exp(\beta G)$ weight (Eq. 4). However, recent work (e.g., BDPO) shows that the standard KL-reg objective (Eq. 2) can be optimized stably via a pathwise KL decomposition, resulting in an analytic MSE penalty. Given this, what is the theoretical advantage of proposing a new "greedified" objective (Eq. 5)?
2. On the Training Mechanism (re: Saturation): The core training relies on $\sigma(\beta G)$ (Eq. 9). This weight saturates for high $G(s,a)$ values. How can the network $\epsilon^+$ learn to distinguish between a "good" action ($G=10$) and an "excellent" action ($G=20$) if the training signal (the weight) is nearly identical for both?
3. On the Inference Mechanism (re: $\omega$): Following Q2, if $\epsilon^+$ has lost the high-reward gradient information due to saturation, how can the inference factor $\omega$ (Eq. 10) recover this information? Is it not simply amplifying a "clipped" or "binary" signal, rather than steering the policy towards the truly optimal (e.g., highest $G(s,a)$) actions? Could the authors comment on what $\epsilon^+$ is actually learning?
4. On the shift factor $k$: Regarding the 'shift factor' $k$ introduced in Appendix D.2 (using $\sigma(\beta G + k)$): This parameter is absent from the theoretical derivation. Could the authors confirm that this is necessary to counteract the sigmoid saturation and center the function's active region over the data's value distribution? How sensitive is the algorithm's performance to the choice of $k$, and doesn't its necessity undermine the robustness and principled nature of the proposed theoretical framework?

---

> ### Author Response · Authors · 2025-11-22
> **Rebuttal to Reviewer ZjGt (1/2)**
>
> We thank the reviewer for the comments on our paper. Regarding the comments of the reviewer, we provide the following responses.
>
> > **W1 & Q1 Relation between DIPOLE and the methods optimizing with decomposing diffusion path(e.g. BDPO, DPPO)**
>
> - First, we did not assume weighted regression is the only way to optimize the KL-regularized RL objective. We choose to develop upon the weighted-regression paradigm primiarly due to its simplicity. In fact, many recent diffusion-based offline RL works follows the same recipe, such as FISOR (ICLR'24)[R1], DTQL (NeurIPS'24)[R2], as well as QVPO (NeurIPS'24)[R3] and DPMD/SDAC (ICML'25)[R4] that are mentioned by **Reviewer jJbf** and **89ky**.
> - The BDPO algorithm mentioned by the reviewer can be considered as an extension of DPPO [R5], i.e., adding a penalty in each diffusion step to implement behavior regularization. Both of the two methods model the denoising process as a multi-step MDP and use Gaussian approximations to compute the likelihood of intermediate denoising steps. However, as we have discussed in Introduction and Preliminary sections, policy optimization through a long diffusion process is very costly and unstable. It requires storing intermediate denoising states during rollout, and leads to substantial memory overhead and considerably more complex training pipeline than standard diffusion model training. Moreover, the crude Gaussian likelihood approxiamtion also suffers from error accumulation. By constrast, our proposed DIPOLE only needs to learn two dichotomous policies using two weighted diffusion loss, which is extremely simple and requires minimal modification to existing diffusion model training pipeline.
> - To further demonstrate the efficiency and superior performance of DIPOLE, we conducted additional experiments on the autonomous driving task, to compare DIPOLE with DPPO and BDPO. We adopt LoRA to RL finetune all the methods, and we kept the same amount of LoRA parameters for fair comparison. All models are trained on a 64x NVIDIA H20 server.(For DPPO and BDPO, we have already tuned learning rate, diffusion discount and finetuned denoising steps for training stability.) The following tables report the training costs and the evaluation scores (PDMS) of different methods. As shown in the results, **DIPOLE has a smaller memory consumption, higher training efficiency, and better PDMS than DPPO and BDPO**.
>
> | Diffusion-RL Algorithm | LoRA Params(M)                  | VRAM(GiB, Total batchsize 2048) | RAM(TiB) | Training Time (10k steps) | PDMS |
> | ---------------------- | ------------------------------- | ------------------------------- | -------- | ------------------------- | ---- |
> | DPPO                   | Single branch 13.4M             | 691                             | 1.04     | 3.6h                      | 89.0 |
> | BDPO                   | Single branch 13.4M             | 691                             | 1.05     | 4.4h                      | 88.6 |
> | DIPOLE                 | Double branch 13.4M (6.7M+6.7M) | **514**                         | **0.25** | **3.2h**                  | 94.1 |

---

> > ### Author Response · Authors · 2025-11-22
> > **Rebuttal to Reviewer ZjGt (2/2)**
> >
> > > **W2 & Q2 & Q3. Confusions on sigmoid saturation**
> >
> > - `Regarding sigmoid saturation`. Although sigmoid function has saturation issue, we'd like to point out that the benefit brought by sigmoid weighting is much larger than its negative impact. Theoretically, it naturally leads to an elegant, decomposable closed-form optimal solution that enables direct policy optimization. Practically, the signmoid weighting scheme leads to strictly bounded weighted regression loss, enable highly stable and efficient diffusion policy learning. Lastly, the saturation issue can also be easily mitigated through proper reward shaping in practice, i.e., adding the $k$ parameter.
> > - `Regarding the reward landscape`. Note in DIPOLE, we simulataneusly learning both the positive policy $\pi^+$ and the negative policy $\pi^-$ through weighting of $\sigma(\beta G)$ and $1-\sigma(\beta G)$. The optimal solution has the form of: $\pi^*(a\mid s)\propto\mu(a\mid s)\sigma(\beta G(s,a))\exp(w\beta G(s,a))=[\pi^+]^{(1+w)}/[\pi^-]^w$ Hence even $\pi^+$ or $\pi^-$ may have saturation weight term, but as the optimal solution is the exponentiated ratio of $\pi^+$ and $\pi^-$, the optimal solution still recovers fine-grained reward landscape due to the contrast of $\pi^+$ and $\pi^-$. Moreover, in our design, $\pi^+$ and $\pi^-$ are oppositely learned, good sample with large $G$ will simulanteously pushes up $\pi^+$ while pushes down $\pi^-$, creating large contrast when computing $[\pi^+]^{(1+w)}/[\pi^-]^w$. What's interesting is that our mechanism exactly mirrors CFG mechanism: $\epsilon_{\mathrm{CFG}}=(1+w)\epsilon_+-w\epsilon_-$. Even if the positive model saturate, the negative model still preserves useful information, and the CFG mechanism amplifies their difference.
> >
> > > **W3 & Q4. Confusions on k shifting**
> >
> > - In our method, $k$ essentially performs reward shaping. Through simple derivation, it can be shown that adding $k$ to $\beta G$ is equivalent to add $k(1-\gamma)/\beta$ to all rewards. Applying reward shaping is a common practical trick in RL, for example, in IQL [R6] as well as most offline RL papers, they typically add $-1$ to all antmaze tasks to ensure reasonable performance. $k$ in our paper is more of a implementation choice rather than part of our theoretical derivation, since it only uniformly shifts the reward for easier numerical optimization, rather than affecting the optimal solution.
> > - Moreover, as we have mentioned previously, adding a $k$ term is a very simple way to mitigate the sigmoid saturation issue as pointed out by the reviewer. In Appendix D.4 Fig. 5 in our paper, we also show that DIPOLE maintains relatively stable performance across different $k$ settings.
> >
> > [R1] Zheng Y, Li J, Yu D, et al. Safe Offline Reinforcement Learning with Feasibility-Guided Diffusion Model. ICLR 2024.
> >
> > [R2] Chen T, et al. Diffusion Policies Creating a Trust Region for Offline Reinforcement Learning. NeurIPS 2024.
> >
> > [R3] Ma H, et al. Efficient Online Reinforcement Learning for Diffusion Policy. ICML 2025.
> >
> > [R4] Ding S, et al. Diffusion-based reinforcement learning via q-weighted variational policy optimization. NeurIPS 2024.
> >
> > [R5] Ren A Z, Lidard J, Ankile L L, et al. Diffusion policy policy optimization. arXiv preprint arXiv:2409.00588, 2024.
> >
> > [R6] Kostrikov I., et al. Offline Reinforcement Learning with Implicit Q-Learning. ICLR 2022.

---

### Official Review · Reviewer_noDV · 2025-11-01

**Soundness:** 3
**Presentation:** 3
**Contribution:** 2
**Rating:** 4
**Confidence:** 3

**Summary:**

This paper introduces DIPOLE (Dichotomous Diffusion Policy Improvement), a stable, scalable, and controllable reinforcement learning framework for training large diffusion policies. Existing RL approaches for diffusion policies either (i) directly optimize value or reward objectives, leading to high gradient variance and training instability, or (ii) approximate Gaussian likelihoods across multiple denoising steps, which are computationally expensive and often inaccurate. By revisiting the KL-regularized RL formulation, DIPOLE proposes a greedified KL-regularized objective that naturally decomposes into two dichotomous sub-policies: the positive policy that favors high-reward actions and the negative policy that models low-reward actions.

**Strengths:**

This work derives a new closed-form optimal policy under a modified KL objective with a bounded sigmoid weighting, effectively avoiding unstable exponential terms and preventing gradient explosions. The dual-policy decomposition enables learning from both high- and low-reward samples, mitigating data imbalance and overfitting to rare high-reward trajectories. Empirical results demonstrate that DIPOLE consistently outperforms strong baselines across offline, offline-to-online, and large-scale VLA tasks. Moreover, the training procedure remains simple and modular, as it only modifies the loss weights of standard diffusion objectives, maintaining full compatibility with existing architectures.

**Weaknesses:**

1.	In DIPOLE, the weight term $\sigma(\beta G(s,a))$ (or $1 - \sigma(\beta G(s,a))$) is treated as constant with respect to the diffusion model parameters. This means that the diffusion model learns to denoise under static weighting but does not explicitly learn how to adjust the action distribution to improve $G(s,a)$ directly. Consequently, there is no gradient signal guiding the modification of intermediate noisy actions to increase the expected reward, which may lead to slower convergence or plateaued performance when the current policy’s support does not already include near-optimal actions. As a result, the performance of DIPOLE heavily depends on the quality of the value estimator. If $G(s,a)$ overestimates certain actions, the weighting will amplify these errors, making policy improvement rely entirely on value accuracy rather than direct reward gradients. This issue is particularly pronounced in offline RL, where value estimates can be severely biased in out-of-distribution (OOD) regions. Thus, the diffusion process learns primarily through denoising consistency instead of reward shaping across time. Furthermore, since updates are based on weighted regression rather than policy gradient optimization, there is no stochastic gradient noise or entropy regularization to encourage exploration. In online fine-tuning scenarios (e.g., DIPOLE’s autonomous driving setup), this lack of exploratory signal could slow adaptation to unseen environments.
2.	The paper derives the optimal policy formulation but does not provide rigorous convergence guarantees or theoretical error bounds for the dichotomous approximation.

3.	Although the parameter $\omega$ controls the degree of greediness, the paper lacks quantitative analysis on how $\omega$ and $\beta$ jointly influence training stability and performance.

4.	Training two separate diffusion models likely doubles computational and memory costs, yet the paper does not report comparisons on training time or efficiency.

5.	The method assumes a reliable reward function or Q-estimator $G(s,a)$, but it remains unclear how performance degrades when the value estimates are noisy or biased.

**Questions:**

1.	For DP-VLA, the reward shaping, return computation, and LoRA-based adaptation are briefly described but not rigorously analyzed.
2.	The paper uses both temperature $\beta$ and greediness $\omega$, how do they jointly affect stability and optimality?
3.	The linear combination resembles CFG, but is $\omega$ chosen adaptively per state or fixed globally? How does this choice affect exploration–exploitation balance?

---

> ### Author Response · Authors · 2025-11-22
> **Rebuttal to Reviewer noDV (1/3)**
>
> We thank the reviewer for the constructive comments on our paper. Regarding the concerns of the reviewer, we provide the following responses.
>
> > **W1 & W5. Confusions on weighted-regression based diffusion optimization methods and the requirement on Q-value estimation quality.**
>
> - `Validity of weighted-regression objective.` First, note that our paper is not the only study that uses weighted-regression objective for diffusion policy optimization. Actually, this is a popular scheme and have already been used in many offline RL studies, such as FISOR (ICLR'24)[R1], DTQL (NeurIPS'24)[R2], as well as QVPO (NeurIPS'24)[R3] and DPMD/SDAC (ICML'25)[R4] that are mentioned by **Reviewer jJbf** and **89ky**. Note that all of these methods uses the non-diffusion-step dependent value term as weight for diffusion policy optimization, and all of them have achieved good performance. This scheme enjoys both simplicity and stability as compared to other diffusion-based policy optimization methods. We refer the reviewer to Theorem 2 of the FISOR paper[R1] and Appendix B.2 of the DPMD/SDAC[R4] paper for the detailed theoretical proofs regarding the validity of such weighted-regression scheme for diffusion policy optimization.
> - `Requirement on the quality of value estimator.` Note that in our RL experiments, we simply adopt the commonly used IQL-style value learning method and the tranditional TD-learning method for all our experiments (see Appendix D.2), without adopting any other more accurate value estimation methods such as distributional Q-value or the more recent flow Q-value[R5], but as we have shown in our experiments, our methods still achieve the best performance among all baselines. These show that our method does not really sensitive to value accuracy as claimed by the reviewer.
> - `Regarding exploration.` Note our paper is primarily focused on **developing a diffusion policy improvement framework**, additional plug-and-play exploration designs such as incorporating exploration noise or incorporating intrinsic rewards can also be seamless integrated in our framework. In fact, due to the great expressivity and inherent multimodal learning ability of diffusion policies, we find simply sampling from it can already produce diverse enough samples for online policy learning even without any extra exploration design.
>
> > **W2. Convergence Guarantees or Theoretical Error Bounds**
>
> Note that we provide closed-form optimal solution of our objective, which is already a very strong theoretical property. This enables directly learning towards the optimal solution instead of relying on some asymptoticaly convergent update schemes. Second, the focus of our paper is to provide a strong empirical algorithm, especially for optimizing large diffusion policies. Deriving detailed theoretical error bounds is out of our scope. Moreover, to ensure our framework is flexibile enough, we allow different forms of return signal $G(s,a)$ and reference policies $\mu$ for diverse offline and online RL task settings, which makes it almost impossible to derive a single theoretical error bound to account for such diverse problem settings.
>
> > **W3 & Q2. Influence of $w$ and $\beta$**
>
> - We thank the reviewer for the suggesetion. We have conducted additional experiments in our revised paper that quantitatively ablation on the joint impact of $w$ and $\beta$ on the final offline RL performance. The results are updated in the Fig. 5 in Appendix D.4. averaged over 4 random seeds. We report some results in the tables below, more results please refer to our revised paper.
> - In general, $\beta$ serves is a reward-shaping coefficient, and $w$ controls the greediness of the combined policy during inference. Larger $\beta$ and larger $w$ typically resulting in more optimisitic policy. But overly large $\beta$ and $w$ may cause instability and reduced performance during policy learning.
>
> **Cheetah-Run**:
> | $w$ \ $\beta$ | 1 | 2 | 4 |
> |----------|-----|-----|-----|
> | 1 | 78+-8 | 106+-15 | 209+-10 |
> | 1.5 | 101+-17 | 121+-14 | 217+-6 |
> | 3 | 123+-18 | 158+-13 | 215+-4 |
> | 5 | 119+-7 | 144+-11 | 205+-3 |
> | 10 | 95+-9 | 114+-22 | 203+-8 |
>
> **Cheetah-Run_backward**:
> | w \ beta | 1 | 2 | 4 |
> |----------|-----|-----|-----|
> | 1 | 72+-14 | 98+-2 | 264+-4 |
> | 1.5 | 87+-7 | 101+-5 | 270+-5 |
> | 3 | 106+-10 | 128+-26 | 255+-41 |
> | 5 | 102+-11 | 128+-6 | 253+-11 |
> | 10 | 105+-19 | 115+-29 | 243+-5 |

---

> > ### Author Response · Authors · 2025-11-22
> > **Rebuttal to Reviewer noDV (2/3)**
> >
> > > **W4 & Q1. Computation and memory costs, as well as other implementation details.**
> >
> > The computation and storage costs of DIPOLE actually not as heavy as the reviewer thinks.
> >
> > - `In terms of computation`, as DIPOLE uses a bounded weighted regression objective, hence learning the dichotomous policies is extremely simple and stable, resulting in fast convergence and efficient training. This forms a sharp contrast with recent DPPO-type of diffusion policy optimization methods [R6, R7], which models the diffusion denoising process as a multi-step MDP, resulting in costly and unstable training.
> > - `In terms of memory cost`, although DIPOLE needs to learn two dichotomous policies, the storage cost can be greatly mitigated by leveraging proper practical implementation. Such as separate LoRA adapters as we have adopted in our DP-VLA experiments, or add a positive/negative indicator as a condition to the same diffusion policy. Such treatments adds negligble amount of extra parameters, without incurring much storage costs.
> > - `Additional experiments on computation/memory costs.` To fully address the concerns from the reviewer, we further conducted additional experiments on the autonomous driving task, to compare DIPOLE with DPPO[R6] and BDPO[R7], these methods treats the diffusion denoising process as a multi-step MDP and use policy gradient to opitmize each intermediate denoising steps. We adopt LoRA to RL finetune all the methods, and we kept the same amount of LoRA parameters for fair comparison. All models are trained on a 64x NVIDIA H20 server. The following tables report the training costs and the evaluation scores (PDMS) of different methods. As shown in the results, **even though DIPOLE learns two dichonomous policies, its training is actually faster and occupies less memory as compared to DPPO-type methods**. The resulting performance of DIPOLE is also greatly outperforms DPPO and BDPO, showing its strength in tuning large diffusion policies.
> >
> > | Diffusion-RL Algorithm | LoRA Params(M)                  | VRAM(GiB, Total batchsize 2048) | RAM(TiB) | Training Time (10k steps) | PDMS |
> > | ---------------------- | ------------------------------- | ------------------------------- | -------- | ------------------------- | ---- |
> > | DPPO                   | Single branch 13.4M             | 691                             | 1.04     | 3.6h                      | 89.0 |
> > | BDPO                   | Single branch 13.4M             | 691                             | 1.05     | 4.4h                      | 88.6 |
> > | DIPOLE                 | Double branch 13.4M (6.7M+6.7M) | **514**                         | **0.25** | **3.2h**                  | 94.1 |
> >
> > - `Other implementation details.`
> >   - For return compuation, in our RL experiments on ExORL and OGBench benchmarks, we use the advantage function $A(s,a)$ as $G(s,a)$. More specifically, to calculate the advantage function, we adopt IQL-style expectile regression loss for value learning in offline RL experiments on ExORL; while on OGBench, we adopt the traditional temporal difference (TD) method for value learning (see Appendix D.2 for details). For the autnomous driving DP-VLA experiments, as it is a single-step RL problem, we directly employed the reward function provided by the NAVSIM benchmark as $G(s,a)$.
> >   - The detail of our LoRA implementation on the DP-VLA experiments are summarized in the table below. We have also included this information in our revised paper.
> >
> > | Hyperparameters                     | Value        |
> > | ----------------------------------- | ------------ |
> > | Rank                                | 16           |
> > | Alpha                               | 16           |
> > | Dropout                             | 0.           |
> > | Num. additional params(Each/Total)  | 6.68M/13.37M |
> > | Ratio additional params(Each/Total) | 1.4%/2.8%    |

---

> > > ### Author Response · Authors · 2025-11-22
> > > **Rebuttal to Reviewer noDV (3/3)**
> > >
> > > > **Q3. Confusions on CFG scale $w$ and exploration–exploitation balance**
> > >
> > > - The parameter $w$ is fixed globally in our experiments, but it is also possible to be adjusted adaptively if we have some prior knowlege of the task, e.g., if we know in some regions, more agressive policy optimization is favorable, then we can use larger $w$. Note that in the offline RL setting, $w$ is not involved during training, only impacting the greediness of policy at inference stage. In the online setting, based on our empirial observation on the DP-VLA, using larger $w$ could increase greediness during online exploration, but being overly greedy also increase the chance to sample on low-return data, reduce learning stability.
> > >
> > > [R1] Zheng Y, Li J, Yu D, et al. Safe Offline Reinforcement Learning with Feasibility-Guided Diffusion Model. ICLR 2024.
> > >
> > > [R2] Chen T, et al. Diffusion Policies Creating a Trust Region for Offline Reinforcement Learning. NeurIPS 2024.
> > >
> > > [R3] Ma H, et al. Efficient Online Reinforcement Learning for Diffusion Policy. ICML 2025.
> > >
> > > [R4] Ding S, et al. Diffusion-based reinforcement learning via q-weighted variational policy optimization. NeurIPS 2024.
> > >
> > > [R5] Agrawalla B, et al. floq: Training Critics via Flow-Matching for Scaling Compute in Value-Based RL. arXiv 2025.
> > >
> > > [R6] Ren A Z, Lidard J, Ankile L L, et al. Diffusion policy policy optimization. arXiv preprint arXiv:2409.00588, 2024.
> > >
> > > [R7] Gao C., et al. Behavior-Regularized Diffusion Policy Optimization for Offline Reinforcement Learning. ICML 2025.

---

### Official Review · Reviewer_89ky · 2025-11-01

**Soundness:** 3
**Presentation:** 4
**Contribution:** 3
**Rating:** 6
**Confidence:** 4

**Summary:**

The paper introduces DIPOLE, a new framework for training diffusion-based policies in goal-conditioned offline reinforcement learning. Instead of using a single diffusion policy with unstable exponential advantage weighting, the authors propose a dichotomous formulation: two separate diffusion policies are trained — one favoring high-return behaviors ($\pi^+$) and one suppressing low-return behaviors ($\pi^-$). The final policy is synthesized at inference by combining their score functions. This approach avoids instability caused by unbounded weighting and enables a controllable trade-off between greediness and safety. Empirically, DIPOLE achieves strong performance across several offline RL tasks (ExORL, OGBench), improves stability in training, and scales to large vision-language-action models in autonomous driving (NAVSIM benchmark), outperforming prior imitation-based diffusion policies.

**Strengths:**

* The paper proposes a simple but effective method (DIPOLE) that trains two diffusion policies instead of one, helping stabilize learning in offline RL.
* The method is well-motivated and theoretically justified, avoiding unstable exponential weighting by using bounded scores.
* Strong experimental results across many tasks, including large-scale vision-language-action models for autonomous driving.
* The paper is clearly written, well-organized, and easy to follow.

**Weaknesses:**

* The method is only evaluated in offline or offline-to-online settings. I am not sure why the same idea can't be applied to online RL?
* The baselines for comparison seem random to me. Not sure what are the reasons to choose those baselines as opposed to some other diffusion-based / non-diffusion-based offline RL baselines. For example, there are plenty of model-based offline RL baselines and I think the authors primarily only choose model-free baselines. Is this intentional? What are the rationals behind choosing these baselines? I have read Sec 4.1 but I am not fully convinced by the explanation.

**Questions:**

Can this algorithm generalize to the online setting? There is some recent work on using KL-regularized RL and mirror descent to define the diffusion policy loss function in the online setting (see below reference), which has the same form as Equation 4. I think the same dichotomous idea may also work there. Could you please explain if this will work or not?

"Efficient Online Reinforcement Learning for Diffusion Policy", Haitong Ma, Tianyi Chen, Kai Wang, Na Li, Bo Dai, ICML 2025

---

> ### Author Response · Authors · 2025-11-22
> **Rebuttal to Reviewer 89ky (1/2)**
>
> We thank the reviewer for the constructive comments on our paper. Regarding the comments of the reviewer, we provide the following responses.
>
> > **W1 & Q1. Application on Online RL.**
>
> - We thank the reviewer for the thoughtful comment. Yes, DIPOLE can also be used in offfline RL, the only modification required is to randomly initialize the base policy and set the reference policy $\mu$ to the policy $\pi_{k-1}$ updated in the last step, akin to trust-region style online policy optimization as TRPO. We have mentioned this in the paragraph below Eq. (2). Actually, our offline-to-online results can be viewed as an online RL fine-tuning setting, the only difference is that we start from a pre-trained diffusion policy instead of a randomly initialized policy.
> - To further demonstrate DIPOLE's applicability to online RL setting, we provide the online RL results on OGbench, as shown in the following table. We remain the hyperparameters the same as the offline-to-online setting, with 1M gradient steps, averaged over 8 random seeds. We find that many scores reported below are actually **higher** than our reported offline-to-online RL results in Table 7 (Appendix D.5), primarily due to the extra conservatism introduced when regularizing the sub-optimal offline pre-trained policy as reference policy in the offline-to-online setting.
>
> | Domain-Task                               | Online Performance |
> | ----------------------------------------- | ------------------ |
> | antsoccer-arena-navigate-singletask-task1 | 96+-3              |
> | antsoccer-arena-navigate-singletask-task2 | 96+-2              |
> | antsoccer-arena-navigate-singletask-task3 | 77+-6              |
> | antsoccer-arena-navigate-singletask-task4 | 74+-6              |
> | antsoccer-arena-navigate-singletask-task5 | 85+-4              |
> | cube-single-play-singletask-task1         | 100+-0             |
> | cube-single-play-singletask-task2         | 100+-0             |
> | cube-single-play-singletask-task3         | 100+-0             |
> | cube-single-play-singletask-task4         | 98+-2              |
> | cube-single-play-singletask-task5         | 99+-2              |
> | cube-double-play-singletask-task1         | 97+-2              |
> | cube-double-play-singletask-task2         | 55+-16             |
> | cube-double-play-singletask-task3         | 71+-10             |
> | cube-double-play-singletask-task4         | 9+-5               |
> | cube-double-play-singletask-task5         | 69+-22             |
> | scene-play-singletask-task1               | 100+-0             |
> | scene-play-singletask-task2               | 98+-4              |
> | scene-play-singletask-task3               | 99+-1              |
> | scene-play-singletask-task4               | 7+-8               |
> | scene-play-singletask-task5               | 0+-0               |
>
> > **W2. Confusions on baselines**
>
> - We thank the reviewer for the comment. In offline RL and offline-to-online RL fields, the current SOTA methods all adopt diffusion/flow matching policies, including the very recent method IFQL and FQL, both are published in 2025. We also compared two other diffusion policy-based RL method IDQL and CFGRL, as the former is a widely used method in diffusion-based RL and the latter is very related to our methods. Method with simpler Gaussian policies currently are not as performant as the more recent diffusion-based methods, but we still choose two widely used offline RL methods, IQL and ReBRAC as our baselines, in order to make our evaluation more comprehensive.
> - Regarding model-based baselines, if the reviewer is familiar with offline RL studies, you will notice that most of the strong offline RL methods are all model-free, and model-based methods are typically underperform model-free methods. In the offline RL setting, rollout data generated from the offline learned models may suffer from serious approximation errors in OOD regions, leading to problematic learning signal, especially when the offline data coverage is limited. This is also discussed in several works [R1, R2]. Therefore, only a few early offline RL works like MOPO[R3], COMBO[R4] adopt model-based design, most of the newer offline RL methods all switch to model-free paradigm. Moreover, many recent strong offline RL methods like IQL, IDQL, FQL (compared in our paper) also only compare with the stronger model-free methods, instead of the less performant model-based methods.
> - Moreover, in large-scale RL settings such as our end-to-end autnomous driving experiments with DP-VLA, adopting a model-based approach requires learning a world model, which can be very difficult to construct and also requires substantially larger training cost when combined in an RL framework.

---

> > ### Author Response · Authors · 2025-11-22
> > **Rebuttal to Reviewer 89ky (2/2)**
> >
> > > Q1. Apply DIPOLE in other online diffusion-based RL algorithms.
> >
> > We thank the reviewer for the suggested reference. Yes, it is possible to extent our dichonomous scheme to DPMD and SDAC in the suggested reference, as they also adopt the similar weighted-regression form. We have cited the suggested reference in our revised paper and will add further discussion in our final version.
> >
> > [R1] Swazinna P, et al. Comparing Model-free and Model-based Algorithms for Offline Reinforcement Learning. IFAC-PapersOnLine, 55(15), 19-26, 2022.
> >
> > [R2] Cheng P, et al. Look beneath the surface: Exploiting fundamental symmetry for sample-efficient offline rl. NeurIPS 2023.
> >
> > [R3] Yu T, et al. Mopo: Model-based offline policy optimization. NeurIPS 2020.
> >
> > [R4] Yu T, et al. Combo: Conservative offline model-based policy optimization. NeuIPS 2021.

---

### Official Review · Reviewer_jJbf · 2025-11-01

**Soundness:** 3
**Presentation:** 3
**Contribution:** 3
**Rating:** 6
**Confidence:** 4

**Summary:**

The paper proposes DIPOLE, a novel reinforcement learning (RL) framework for optimizing diffusion-based policies. The key idea is to reformulate the KL-regularized RL objective into a “greedified” version that can be decomposed into two dichotomous diffusion policies — one maximizing reward (positive policy) and one minimizing reward (negative policy). During inference, their score functions are linearly combined, enabling controllable trade-offs between greediness and stability.
Extensive experiments on ExORL, OGBench, and a large-scale autonomous driving benchmark (NAVSIM) demonstrate performance gains over prior RL and diffusion-based baselines.

**Strengths:**

1. The proposed dichotomous decomposition of the KL-regularized objective is both elegant and conceptually novel. The analogy to classifier-free guidance (CFG) provides a strong intuitive and theoretical bridge between diffusion modeling and RL optimization.

2. The paper presents comprehensive experiments across multiple RL benchmarks and an ambitious large-scale 1B-parameter VLA model for end-to-end driving, showing clear improvements over strong baselines (IQL, FQL, CFGRL, etc.).

3. The paper presents comprehensive experiments across multiple RL benchmarks and an ambitious large-scale 1B-parameter VLA model for end-to-end driving, showing clear improvements over strong baselines (IQL, FQL, CFGRL, etc.).

**Weaknesses:**

1. The reviewer is a little bit confused about why we need to train a policy that minimizes the rewards. In my opinion, to avoid the large difference between the optimized policy and the behavior policy of offline data, we can directly perform imitation learning on the second diffusion policy rather than minimizing the reward.
2. How can we get $G(s, a)$ in the proposed method? Should we apply some special technique to learn it, such as CQL [R2]?
3. The method can be classified as a weighted-based diffusion RL method and lacks the citation of the recent weighted-based diffusion RL method [R1].

[R1] Ding S, Hu K, Zhang Z, et al. Diffusion-based reinforcement learning via q-weighted variational policy optimization[J]. Advances in Neural Information Processing Systems, 2024, 37: 53945-53968.

[R2] Kumar A, Zhou A, Tucker G, et al. Conservative q-learning for offline reinforcement learning[J]. Advances in neural information processing systems, 2020, 33: 1179-1191.

**Questions:**

The proposed method requires training two separate policy networks $\epsilon^+, \epsilon^-$, which effectively doubles the computational and storage costs. However, the paper does not discuss or evaluate this overhead: How long is the training time compared to single-policy baselines? During inference, although the final score is obtained by a linear combination, two models must be executed—what is the resulting latency? In the autonomous driving experiments, the authors mention using LoRA to mitigate this issue, but they do not quantify the number of additional parameters introduced by LoRA or its impact on training efficiency

---

> ### Author Response · Authors · 2025-11-22
> **Rebuttal to Reviewer jJbf (1/2)**
>
> We thank the reviewer for the constructive comments on our paper. Regarding the comments of the reviewer, we provide the following responses.
>
> > **W1.The necessity of negative policy**
>
> - `A Direct Consequence of Our Objective.` The optimal solution of our proposed greedified KL-regularized RL objective in **Theorem 1** naturally decomposes into a positive and a negative policy. If we replace the negative policy with an imitation policy, the combined policy satisfies $\hat{\pi}(a\mid s)\propto\mu(a\mid s)\sigma(\beta G(s,a))^{w+1}$, which deviates from the theoretically derived optimal solution $\pi^*(a\mid s)\propto\mu(a\mid s)\sigma(\beta G(s,a))\exp(w\beta G(s,a))$ in **Theorem 1**.
> - `Benefits of Learning Negative Policy.` Second, learning a negative policy also introduces a set of benefits. Learning the negative policy enables fully utilizing bad data for policy optimization, helps suppress bad behaviors and drive the final combined policy away from sub-optimal behaviors. It also resolves the issue of policy learning been dominated by high-return samples as in exp-weighted regression, and enabling more efficient learning, as we have discussed in Section 3.2,
> - `Emprirical Performance.` To further address the concern of the reviewer, we have conducted additional experiments on ExORL benchmark (averaged on 8 random seeds) and the autonomous driving DP-VLA model. We replace the negative policy $\pi^-$ in DIPOLE with the imitation policy $\mu$. We observe that **using imitation policy $\mu$ as negative policy results in slower convergence and less greedy optimization**. The results are presented in the following table:
>
> | Domain-Task                     | Imitation $\mu$ | Negative $\pi^-$ |
> | ------------------------------- | ------ | ------ |
> | Walker-Stand                    | 903+-6 | 953+-4 |
> | Walker-Walk                     | 894+-7 | 910+-5 |
> | Walker-Run                      | 432+-9 | 442+-9 |
> | Cheetah-Run                     | 271+-11 | 274+-12 |
> | Cheetah-Run-backward            | 341+-13 | 350+-15 |
> | Jaco-Reach-top-right            | 95+-26 | 119+-23 |
> | Jaco-Reach-top-left             | 92+-13 | 113+-8 |
>
> For DP-VLA experiment, we present the performance for training 10k steps. More detailed training curves can be found in our revised paper.
>
> | Algorithm         | PDMS  |
> | ----------------- | ----- |
> | DIPOLE(Origin)    | **94.1** |
> | DIPOLE(Imitation) | 93.1 |
>
> > **W2. How to get proposed function $G(s,a)$?**
>
> - As we have described in the begining of Section 3.1, our proposed DIPOLE is a very flexible framework, which allows $G(s,a)$ taking forms of either the reward function as in single-step RL problems like LLM RLFT, or value $Q(s,a)$ or advantage function $A(s,a)$ as in standard multi-step settings.
> - For our RL experiments on ExORL and OGBench benchmarks, we use the advantage function $A(s,a)$ as $G(s,a)$. More specifically, to calculate the advantage function, we adopt IQL-style expectile regression loss for value learning in offline RL experiments on ExORL; while on OGBench, we adopt the traditional temporal difference (TD) method for value learning (see Appendix D.2 for details). For the autnomous driving DP-VLA experiments, as it is a single-step RL problem, we directly employed the reward function provided by the NAVSIM benchmark as $G(s,a)$.

---

> > ### Author Response · Authors · 2025-11-22
> > **Rebuttal to Reviewer jJbf (2/2)**
> >
> > > **W3. Citations of recent weighted-based diffusion RL method [R1].**
> >
> > We thank the reviewers for providing this relavent reference. In our initial submission, we have cited FISOR[R2], which is pretty similar to [R1], but appears earlier in literature and also provides theoretical proofs for weighted-based diffusion RL. We have updated our paper to include the suggested reference, as well as several other relevant follow-up works.
> >
> > > **Q1. Computational and storage costs in practical implementations.**.
> >
> > The computation and storage costs of DIPOLE actually not as heavy as the reviewer thinks.
> >
> > - `In terms of computation`, as DIPOLE uses a bounded weighted regression objective, hence learning the dichotomous policies is extremely simple and stable, resulting in fast convergence and efficient training. This forms a sharp contrast with recent DPPO-type of diffusion policy optimization methods [R4, R5, R6], which models the diffusion denoising process as a multi-step MDP, resulting in costly and unstable training.
> > - `In terms of storage cost`, although DIPOLE needs to learn two dichotomous policies, the storage cost can be greatly mitigated by leveraging proper practical implementation. Such as separate LoRA adapters as we have adopted in our DP-VLA experiments, or add a positive/negative indicator as a condition to the same diffusion policy. Such treatments adds negligble amount of extra parameters, without incurring much storage costs.
> > - `Additional experiments on computation/storage costs.` To fully address the concerns from the reviewer, we have conducted additional experiments on the end-to-end autonomous driving task during rebuttal, to compare DIPOLE with recent diffusion-RL methods like DPPO[R4] and BDPO[R5]. We adopt LoRA to RL finetune all the methods, and we kept the same amount of LoRA parameters for fair comparison. All models are trained on a 64x NVIDIA H20 server. The following tables report the training costs and the evaluation scores (PDMS) of different methods. As shown in the results, **even though DIPOLE learns two dichonomous policies, its training is actually faster and occupies less memory as compared to DPPO-type methods**. The resulting performance of DIPOLE is also greatly outperforms DPPO and BDPO, showing its strength in tuning large diffusion policies.
> >
> > | Diffusion-RL Algorithm | LoRA Params(M)                  | VRAM(GiB, Total batchsize 2048) | RAM(TiB) | Training Time (10k steps) | PDMS |
> > | ---------------------- | ------------- | ------------------------------- | -------- | ------------------------- | ---- |
> > | DPPO                   | Single branch 13.4M             | 691                             | 1.04     | 3.6h                      | 89.0 |
> > | BDPO                   | Single branch 13.4M             | 691                             | 1.05     | 4.4h                      | 88.6 |
> > | DIPOLE                 | Double branch 13.4M (6.7M+6.7M) | **514**                         | **0.25** | **3.2h**                  | 94.1 |
> >
> > - `LoRA implementation and inference Latency.` The detail of our LoRA implementation on the DP-VLA experiments are summarized in the table below.
> >
> > | Hyperparameters                     | Value        |
> > | ------- | ---- |
> > | Rank                                | 16           |
> > | Alpha                               | 16           |
> > | Dropout                             | 0.           |
> > | Num. additional params(Each/Total)  | 6.68M/13.37M |
> > | Ratio additional params(Each/Total) | 1.4%/2.8%    |
> >
> > The following table reports the inference time on single and dual LoRA branch in DP-VLA experiment. We also tested different amount of LoRA parameters on the double branch implemnetation. All inference time results are meatured on 1x NVIDIA H20. It shows that using two LoRA branches do increase the inference latency, but within an acceptable range (1s vs 2s) in practice. On the other hand, different amount of the LoRA parameter sizes on our double branch implementation do not have noticeable impact on inference latency.
> >
> > | Config                                                | Time(s) |
> > | ---- | ------- |
> > | Single Branch with LoRA(13.4M)                        | 0.84    |
> > | Double Branch with LoRA(1.7M)                         | 2.02    |
> > | **Double Branch with LoRA(13.4M, used in the paper)** | 1.97    |
> > | Double Branch wtih LoRA(53M)                          | 2.01    |
> >
> > ### References
> >
> > [R1] Ding S, Hu K, Zhang Z, et al. Diffusion-based reinforcement learning via q-weighted variational policy optimization. NuerIPS 2024.
> >
> > [R2] Zheng Y, Li J, Yu D, et al. Safe Offline Reinforcement Learning with Feasibility-Guided Diffusion Model. ICLR 2024.
> >
> > [R4] Ren A Z, Lidard J, Ankile L L, et al. Diffusion policy policy optimization. arXiv preprint arXiv:2409.00588, 2024.
> >
> > [R5] Gao C., et al. Behavior-Regularized Diffusion Policy Optimization for Offline Reinforcement Learning. ICML 2025.
> >
> > [R6] Black K, et al. Training diffusion models with reinforcement learning. ICLR 2024.

---

### Author Response · Authors · 2025-11-22
**General Response**

We thank all the reviewers for the effort engaged in the review phase and the constructive comments.

**We have revised our paper (highlighted in blue text color). The modifications are summarized as follows.**

1.(For Reviewer 89ky) We add the results of FQL and IFQL on ExORL bench in Section 4.1.

2.(For Reviewer jJbf and 89ky) We add citations of recent weighted-based diffusion RL method in Section 1 and 5.

3.(For Reviewer jJbf and noDV) We add further details on the LoRA adapters of DP-VLA experiment in Appendix E.4.

4.(For Reviewer noDV) We add an ablation study on $w$ and $\beta$ in Appendix D.4.

5.(For Reviewer ZjGt) We add the results of DPPO and BDPO on DP-VLA in Table 4 and Appendix D.5

---

### Author Response · Authors · 2025-11-27
**A Kindly Reminder: Looking Forward to Further Discussicon**

Dear reviewers of paper 18486,

As the discussion period is coming to a close, we would be happy to clarify further, and grateful for any other feedback you may povide. We really appreciate your time engaged in the review and rebuttal phase.

Thank you very much and look forward to your replies!

Best regards,

Authors of Paper 18486

---

### Author Response · Authors · 2025-12-03
**Summary for AC (1/2)**

We sincerely appreciate the time and valuable feedback provided by the reviewers during the review stage. In light of the current situation, we summarize the key information regarding DIPOLE—including the paper’s contributions, the main reviewer comments, and our corresponding rebuttals, to facilitate a quick and clear understanding of the paper’s status for the AC.

&nbsp;

---

# Main Contributions and Revisions

&nbsp;

DIPOLE (Dichotomous Diffusion Policy Improvement) is a RL framework for stable, controllable and scalable diffusion-based policy optimization, which has the following 3 characteristics:

- `Stability`: Splits the unstable exponential weighting into two **bounded** components to enable **stable** diffusion-policy learning.

- `Controllability`: Reconstructs the final policy via a linear combination of the two **dichotomous policies’ scores**, yielding **CFG-like greediness control**.

- `Scalability`: Outperforming strong baselines across **offline**, **offline-to-online**, and **large-scale VLA** tasks.

During the rebuttal period, we supplemented: 1) RL benchmarks with additional baselines and ablations, 2) provided detailed implementation information for the autonomous driving experiments, 3) added performance comparisons with other diffusion-based RL methods, and 4) included new experiments and analyses on both training and inference computational costs.

---

# Summary of Reviewer Comments and Responses

&nbsp;

# **Reviewer jJbf**


`Positive Feedback` A strong intuitive and theoretical bridge between diffusion modeling and RL optimization.

### **Main Concerns Q&A**

`Q: Why not directly perform imitation learning on the second diffusion policy rather than minimizing the reward?` A: We provide a theoretical analysis of the differences between the respective closed-form solutions and present **benchmark performance comparisons** for the two approaches.

`Q: The computational and memory overhead.` A: We provide **detailed analyses of computational and memory costs**, and compare them against other diffusion RL methods(DPPO/BDPO), where **DIPOLE achieves substantially superior performance**.

---

&nbsp;

# **Reviewer 89ky**


`Positive Feedback` DIPOLE is a simple, well-motivated, theoretically grounded dual-diffusion approach that stabilizes offline RL, delivers strong results across diverse tasks (including large-scale VLA driving), and is clearly written and easy to follow.


### **Main Concerns Q&A**

`Q: DIPOLE's implementation on Online RL.` A: DIPOLE’s online viability is demonstrated by its offline-to-online results and **further validated its online performance on OGbench**, where it continues to show **stable and strong performance**.

`Q: Choice of baselines and why not including MBRL methods. `A: We explain that MBRL baselines are excluded because **they face severe OOD issues**, incur **high costs** for large-scale tasks, and are generally absent from recent comparable evaluations.

---

> ### Author Response · Authors · 2025-12-03
> **Summary for AC (2/2)**
>
> ## **Reviewer noDV**
>
> `Positive Feedback` The reviewer believes that the bounded closed-form KL policy and dual-policy design make DIPOLE stable, effective across diverse tasks, and easy to train while remaining fully compatible with existing diffusion architectures.
>
> ### **Main Concerns Q&A**
>
> `Q: The validity of weighted-regression.` A: Weighted regression is **standard and theoretically supported** in diffusion RL. We argue that DPPO/BDPO’s multi-step-MDP formulation is costly and unstable, and show through additional autonomous-driving experiments that **DIPOLE is more memory-efficient, easier to train, and achieves better performance**.
>
> `Q: The convergence and error bounds.` A: DIPOLE already has a closed-form solution, so no convergence or error-bound proof is required.
>
> `Q: The influence of hyperparameters $w$ and $\beta$.` A: We add additional ablations on $w$ and $\beta$. And it shows stable and predictable performance.
>
> `Q: Computation and memory costs, other implementation details.` A: DIPOLE offers clear computational and memory advantages over multi-step MDP diffusion methods like DPPO/BDPO.
>
> `Q: Confusions on CFG scale $w$ and exploration–exploitation balance` A: $w$ is a global constant that controls the exploration–exploitation balance.
>
> ---
>
> &nbsp;
>
> ## **Reviewer ZjGt**
>
> `Positive Feedback` The reviewer believes the work is original in its reward-maximizing/minimizing policy decomposition and is strongly validated by impressive empirical results, including scalable fine-tuning of a 1-billion-parameter VLA model.
>
> ### **Main Concerns Q&A**
>
> `Q: DIPOLE is more complex than the methods optimizing with decomposing diffusion path(DPPO/BDPO style).` A: DIPOLE is shown to be more **stable, efficient, and theoretically grounded** than DPPO/BDPO, achieving better performance with **lower time and memory cost**.
>
> `Q: Sigmoid saturation causes a complete loss of gradient information.` A: The sigmoid naturally follows from the closed-form solution, and the negative policy with CFG fully recovers the reward landscape, with saturation mitigated by simple reward shaping.
>
> `Q: Confusions on k shifting.` A: We clarify that k-shifting is just a standard reward-shaping technique.

---

### Meta-Review · Area_Chair_P2LW · 2026-01-07

**Summary:**

Across the four reviews, there is broad agreement that the paper’s core idea is novel and practically impactful: DIPOLE reformulates a KL-regularized objective into a “greedified” form that yields two diffusion policies (positive/negative), and combines their scores at inference in a CFG-like way. Reviewers also consistently acknowledge strong empirical performance, including ambitious scaling to a 1B-parameter VLA driving setting. The suggested decision is primarily shaped by the following concerns:

• Soundness / necessity of the reformulation: One reviewer argues the motivation (“standard KL objective is unusable”) is overstated because DPPO/BDPO-style pathwise KL can optimize the standard objective stably; hence DIPOLE might be an unnecessary detour.

• Training signal / sigmoid saturation and theory–practice gap: There are concerns that bounded sigmoid weights may clip preference information (good vs. excellent), and that the practical “shift” used in the implementation suggests sensitivity not captured in the main theoretical story.

• Dependence on value/Q estimation in offline RL: Questions remain about robustness when critics are biased/noisy (especially OOD), since the method is still weighted-regression driven.

• Compute and latency overhead of two policies: Multiple reviewers asked for explicit training-time and inference-latency accounting and mitigation strategy quantification.

**Reviewer Concerns:**

1. Reviewer jJbf (score 6):

Mostly Addressed:

•  Why train a negative policy? The rebuttal provides (i) a theory-based argument (“direct consequence of the derived optimal solution”) and (ii) empirical evidence showing replacing the negative policy with imitation degrades convergence/performance on ExORL and slightly on DP-VLA.

•  How to obtain the return signal r/ advantage: The rebuttal clearly specifies how ris instantiated in each benchmark (advantage via IQL-style value learning, TD on OGBench, reward for NAVSIM).

•  Compute/storage overhead and LoRA details: The rebuttal provides concrete numbers for VRAM/RAM/training time and LoRA parameter counts, and includes inference latency measurements.

Partially Addressed:

•  Positioning w.r.t. “weighted diffusion RL” literature: They added the requested citation and related works, which helps. Still, the paper’s conceptual novelty relative to that family may remain mildly unclear to this reviewer, but this is more “framing” than a fatal issue.

2. Reviewer 89ky (score 6)

Mostly Addressed:

•  Online RL applicability: The rebuttal claims DIPOLE extends naturally to online RL (reference policy = previous policy) and provides an online OGbench table with results.

•  Baseline selection skepticism: The rebuttal gives a coherent rationale: diffusion/flow policies are current SOTA; includes both diffusion and classic baselines; model-based offline RL typically underperforms due to OOD model error and is impractical at VLA scale.

Partially Addressed:

•  The baseline justification is better, but the “random baselines” concern may not be fully eliminated unless the camera-ready explicitly clarifies selection criteria and possibly adds one or two “representative” missing baselines (if feasible). This is minor.

Reviewer noDV (score 4)

Mostly Addressed:

•  Compute/memory costs: The rebuttal provides quantitative cost comparisons vs DPPO/BDPO and argues bounded regression is simpler and faster.

•  α/β interaction: They added ablations and summarized trends; this directly answers the request for quantitative analysis.

•  CFG scale choice: They clarify βis fixed globally in experiments and discuss potential adaptive use; they explain offline vs online roles.

Partially Addressed Still:

•  Core methodological concern: “weights treated as constant ⇒ no direct reward gradient; heavy reliance on critic accuracy.” The rebuttal largely argues that many prior methods do this and that empirically it works even with standard value learning. It does not fully address the deeper question of failure modes under critic bias / limited support, nor does it provide a targeted robustness study (e.g., controlled critic noise, OOD stress test). This remains the most substantive unresolved point from this reviewer’s perspective.

•  Lack of convergence/error bounds: The rebuttal essentially says out-of-scope; that is acceptable for many ML venues, but it won’t convert a skeptical reviewer on “soundness.”

Reviewer ZjGt (score 4)

Mostly Addressed:

•  DPPO/BDPO comparison: The rebuttal explains the design choice (simplicity, avoiding multi-step MDP over denoising, memory overhead, Gaussian approximation errors) and adds a quantitative comparison showing better efficiency and PDMS in their driving setup.

•  Shift factor k: They argue it is reward shaping (uniform shift) and common in RL practice, and show stability across settings.

Partially Adressed:

•  “Unsound premise / unnecessary objective” critique: The rebuttal reframes the motivation (not “only way,” but “simplicity/efficiency”), which helps, but the reviewer’s claim is about principled necessity: if standard objective can be optimized stably (BDPO), what is the theoretical advantage of changing the objective?

•  Sigmoid saturation critique: The rebuttal claims the ratio of positive/negative policies restores fine-grained structure and analogizes to CFG, and says kmitigates saturation. This is a reasonable story, but it doesn’t fully rebut the reviewer’s core point that saturation may collapse distinctions in the high-return regime. Without a more direct analysis (e.g., demonstrating non-binary weighting distribution, or sensitivity to scaling / saturation thresholds), this concern may remain for that reviewer.

**Reviewer Scores:**

jJbf: +1

89ky: +1

noDV: +1

ZjGt: maybe no change.

---

### Decision · Program_Chairs · 2026-01-26

Accept (Poster)